# Mitigating Disparate Impact of Differentially Private Learning through Bounded Adaptive Clipping

## Abstract

Differential privacy (DP) has become an essential framework for privacy-preserving machine learning. Existing DP learning methods, however, often have disparate impacts on model predictions, e.g., for minority groups. Gradient clipping, which is often used in DP learning, can suppress larger gradients from challenging samples. We show that this problem is amplified by adaptive clipping, which will often shrink the clipping bound to tiny values to match a well-fitting majority, while significantly reducing the accuracy for others. We propose bounded adaptive clipping, which introduces a tunable lower bound to prevent excessive gradient suppression. Our method improves worst-class accuracy by over 10 percentage points on Skewed and Fashion MNIST compared to unbounded adaptive clipping, 7 points compared to Automatic clipping, and 5 points compared to constant clipping.

## 1 Introduction

Differential privacy (DP; Dwork et al. 2006b; Dwork & Roth 2014) is a widely accepted framework for preserving privacy in data analysis, including during machine learning model training. While mitigating privacy issues, DP can exacerbate disparate impact problems (Bagdasaryan et al., 2019; Fioretto et al., 2022; Petersen et al., 2023). The current state-of-the-art (SOTA) solution to address these issues in differentially private stochastic gradient descent (DP-SGD) is based on using adaptive clipping to reduce disparate impacts for minority and confusable groups, groups that share strong similarities in their features (Esipova et al., 2023). However, as we demonstrate in this work, the current methods (Andrew et al., 2021; Esipova et al., 2023) can actually suppress gradients from these groups when dynamically adjusting the clipping bounds, leading to very biased estimates and class-wise disparities due to decreased worst-class performance (see Figure 1).

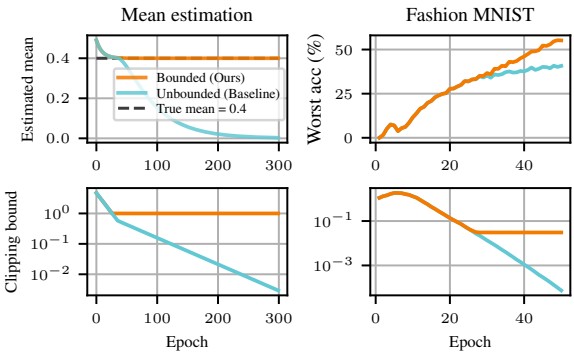

Figure 1: Existing adaptive clipping methods can lead to vanishing clipping bounds (blue), resulting in severe performance degradation for minorities and challenging examples. Setting a lower bound for clipping bound (orange) rectifies this. Left: Mean estimation in bimodal data can converge to the mean of the majority ignoring the minority, Right: The training trajectory of a convolutional neural networks with DP on Fashion MNIST shows significant impact in most difficult class accuracy.

To address the issue, we propose **lower-bounded adaptive clipping**, a mechanism aimed at mitigating the limitations of unbounded adaptive clipping. By introducing a tunable lower bound, our method preserves critical gradient updates for minority and confusable groups while maintaining formal DP guarantees. We evaluate the performance of our method under both non-DP and DP hyperparameter optimization (HPO, Liu & Talwar 2019; Papernot & Steinke 2022), and demonstrate its efficiency in comparison to the current SOTA as well as robustness to HPO stochasticity across diverse datasets and model architectures.

**Related work** Disparate impact as a formal metric has received significant attention in machine learning, with various formulations proposed (Dwork et al., 2012; Kusner et al., 2017; Corbett-Davies et al., 2017). Ensuring low disparate impact becomes even more challenging when combined with the complexities of DP. Recent work has highlighted that DP can disproportionately degrade the performance of minority or confusable groups, making the mitigation of such accuracy disparities a central concern in private learning (Bagdasaryan et al., 2019; Fioretto et al., 2022). In deep learning, accuracy parity, defined as achieving similar accuracy across all demographic or label groups, is considered an important metric in the realm of fairness analysis, and it is especially sensitive to data imbalance and algorithmic design (Tran et al., 2021).

To address these challenges through the lens of disparate impact, one key research direction in recent work has explored improving clipping mechanisms in DP optimization. Starting from DP-SGD with gradient clipping (Abadi et al., 2016), Tran et al. (2021) examined how constant clipping and loss re-weighting influence accuracy disparity across groups, offering insights into the disparate-impact behavior of DP-trained models. Xu et al. (2021) proposed DP-SGD-Fair, which sets group-specific clipping bounds based on sample sizes and adjusts the noise levels accordingly.

Looking again at the overall performance, Bu et al. (2023) proposed automatic clipping (AUTO), where the clipping bound is set individually for each sample, close to its gradient norm. This removes the need for tuning, but effectively rescales all updates and can disadvantage subgroups with large gradient norms. Andrew et al. (2021) introduced an adaptive clipping mechanism that tracks a specific quantile of gradient norms under DP, resulting in a threshold multiplier that depends on the data distribution. The convergence properties of this method were later analyzed by Shulgin & Richtárik (2024), who provided theoretical guarantees on its performance and utility. To better address the disparate impact, Esipova et al. (2023) proposed an adaptive parameterization for clipping bound updates to mitigate misalignment issues in earlier approaches.

In this work, in addition to the standard constant clipping in DP (Abadi et al., 2016), we use the adaptive clipping methods (Andrew et al., 2021; Esipova et al., 2023) and automatic clipping (Bu et al., 2023) as SOTA baselines to evaluate our approach.

**Contributions** Our paper makes the following contributions:

1. We identify a common failure mode with the current SOTA adaptive clipping methods leading to vanishing clipping bounds, resulting in performance and disparate-impact issues (Figure 1). To address these issues, we propose a novel lower-bounded adaptive clipping method that introduces a tunable lower bound to protect minority and confusable groups (Section 3).
2. We evaluate the performance of our proposed approach comprehensively across four datasets and three models, showing its ability to achieve both strong overall performance and accuracy parity under DP constraints compared to existing methods (Section 4.2).
3. We test the performance of our method under differentially private hyperparameter optimization (DP-HPO), demonstrating that our approach is robust to DP-HPO stochasticity compared to existing methods (Section 4.3).

## 2 PRELIMINARIES

**Differential privacy (DP)** DP (Dwork et al., 2006b; Dwork & Roth, 2014) is a mathematical framework for privacy preservation, centered on the principle of quantifying privacy through the comparison of output probabilities between adjacent datasets, formalized as follows.

**Definition 2.1.** *(Approximate DP; Dwork et al., 2006b;a) A stochastic algorithm $\mathcal{M} : \mathcal{D} \to \mathcal{R}$ is $(\varepsilon, \delta)$-DP if for any adjacent datasets $D, D' \in \mathcal{D}$, and for any $S \in \mathcal{R}$, it holds that*

$$\Pr[\mathcal{M}(D) \in S] \le e^{\varepsilon}\Pr[\mathcal{M}(D') \in S] + \delta.$$

In this work, we use sample-level add/remove adjacency, so $D$ and $D'$ are adjacent, if $D$ can be turned into $D'$ by adding or removing a single sample.

**Differentially private stochastic gradient descent (DP-SGD)** To incorporate DP into deep learning, a common approach is to use DP-SGD for optimization. DP-SGD extends SGD with $\ell_2$ norm

gradient clipping and noise injection (Song et al., 2013; Abadi et al., 2016). In effect, clipping bounds the influence any single sample can have on the outcome, after which calibrated Gaussian noise is added to the clipped per-sample gradients to guarantee DP (Dwork et al., 2006a).

However, the update magnitude in standard DP-SGD is influenced by two hyperparameters, the learning rate and the clipping bound, which both affect the update magnitude. Since this interdependence complicates hyperparameter tuning, De et al. (2022) proposed normalizing the learning rate by scaling all gradients by a factor of $\frac{1}{C}$, where $C$ is the clipping bound (see Algorithm 1). This normalization decouples the learning rate and the clipping bound so the hyperparameter $C$ exclusively controls the clipping bound without affecting the update magnitude, simplifying HPO.

---

**Algorithm 1** Normalized DP-SGD (De et al., 2022)

**Input:** Iterations $T$, dataset $D$, sampling rate $q$, expected batch size $B = qN$, clipping bound $C$, noise multiplier $\sigma$, loss function $\mathcal{L}$, initial parameters $\theta_0$ of model $f$.
**for** iteration $t = 0, 1, \ldots, T - 1$ **do**
    $\mathcal{B}_t \leftarrow$ Poisson subsample of $D$ with rate $q$
    **for** $(x_i, y_i) \in \mathcal{B}_t$ **do**
        $g_i \leftarrow \nabla\mathcal{L}(f_{\theta_t}(x_i), y_i)$
        $\bar{g}_i \leftarrow g_i \cdot \min(\frac{1}{C}, \frac{1}{||g_i||})$
    **end for**
    $\tilde{g}_t \leftarrow \frac{1}{B} \left( \sum_{i \in \mathcal{B}_t} \bar{g}_i + \mathcal{N}(0, \sigma^2\mathbf{I}) \right)$
    $\theta_{t+1} \leftarrow \text{OptimizerUpdate}(\theta_t, \tilde{g}_t)$
**end for**

---

In the rest of this paper, DP-SGD refers to DP-SGD with normalization. We note that due to the standard post-processing properties of DP (Dwork & Roth, 2014), any optimizer with access only to the DP gradients, e.g., Adam (Kingma & Ba, 2015), will also satisfy DP. We therefore use the general OptimizerUpdate in Algorithms 1 and 2 to refer to any such optimizer.

**Differentially private hyperparameter optimization (DP-HPO)** Finding good hyperparameters is critical for ensuring good performance, yet finding them especially under DP constraints is non-trivial due to the high computational cost of DP training (Koskela & Kulkarni, 2023) and the risk of extra privacy leakage from HPO (Liu & Talwar, 2019). Papernot & Steinke (2022) have analyzed DP-HPO procedures, showing that privacy leakage can remain modest as long as each training run adheres to DP guarantees.

Considering the intersection of disparate impact and DP which is the focus of this work, the minority and confusable groups are often most at risk from privacy breaches (Xu et al., 2021). Hence, accounting the privacy budget throughout the entire DP pipeline including HPO can be useful to assert the risk. However, as DP-HPO introduces additional randomness into the hyperparameters, it becomes more important to evaluate the robustness of any method to such stochasticity.

## 3 ADAPTIVE CLIPPING ALGORITHMS FOR DP-SGD

While Andrew et al. (2021) first proposed adaptive clipping in the context of DP, Esipova et al. (2023) introduced mechanisms specifically designed to mitigate the disparate impact on different groups. The two algorithms are both special cases of a more general unified unbounded adaptive clipping algorithm, formalized in Algorithm 2, with $C_{\text{LB}} = 0$. This algorithm reduces to that of Andrew et al. (2021) when $\tau = 1$ and to that of Esipova et al. (2023) when $\eta_C = 1$.

### 3.1 KEY HYPERPARAMETERS

Adaptive clipping involves several hyperparameters that control its behavior and effectiveness. Below, we detail their roles and highlight key observations from prior work and our analysis.

**Target quantile** ($\gamma$) specifies the proportion of gradients with norms exceeding the current threshold that the algorithm aims for (Andrew et al., 2021). The clipping bound update will converge exponentially towards a bound where fraction $1 - \gamma$ of the gradients are clipped. The value of $\gamma$ is directly linked to accuracy disparity, as fraction $1 - \gamma$ of the gradients are ignored when considering the clipping bound. The specific behaviour observed in the toy model of Figure 1 (left) could be avoided by setting $\gamma$ to be sufficiently larger than $0.6$ to also consider the minority. Things get more difficult when the size of the minority group is unknown and when it is small, because the estimation of extreme quantiles of the gradient norm distribution is less reliable, so simply using $\gamma \approx 1$ is not a

silver bullet. The optimal value of $\gamma$ is tightly coupled to the threshold multiplier $\tau$ that we discuss next.

**Threshold multiplier** ($\tau$) for counting clipped gradients, introduced by Esipova et al. (2023), is a multiplier that determines the upper limit for identifying outlier gradient norms. Gradients with norms exceeding $\tau \cdot C_t$, where $C_t$ is the clipping bound at the current iteration $t$, are treated as outliers and contribute to updating the clipping bound such that their fraction should become approximately $\gamma$. In practice, $\tau$ and $\gamma$ are tightly coupled. Under a fixed gradient norm distribution, the same fixed point for clipping bound adaptation could be reached by changing $\tau$ and $\gamma$ together suitably. Interestingly, both $\gamma$ and $\tau$ were very stable in our experiments and the values $\gamma = 0.5$ and $\tau = 2.5$ were optimal for all datasets used in the experiments.

The other hyperparameters are similar to DP-SGD. **Initial clipping bound** ($C_0$) serves as the initial value for the adaptive clipping mechanism. **Clipping bound learning rate** ($\eta_C$) defines the learning rate for the updates of the adaptive clipping bound. **Noise multiplier for clipped gradient counting** ($\sigma_{\text{count}}$) determines the scale of noise added to the privatized estimate of how many gradients exceed the current clipping bound. A larger $\sigma_{\text{count}}$ gives more privacy budgets to the counting mechanism, but may result in less accurate estimates, potentially affecting the stability of the adaptive clipping updates. Following Andrew et al. (2021) we use $\eta_C = 0.2$ and following Esipova et al. (2023) we set $\sigma_{\text{count}} = 10\sigma_{\text{grad}}$.

---

**Algorithm 2** Unified normalized DP-SGD with adaptive clipping mechanism (unbounded / lower-bounded)

---

**Input:** Iterations $T$, dataset $D$, sampling rate $q$, expected batch size $B = qN$, initial clipping bound $C_0$, noise multiplier for gradients $\sigma_{\text{grad}}$, noise multiplier for clipped gradient count $\sigma_{\text{count}}$, loss function $\mathcal{L}$, initial parameters $\theta_0$ of model $f$, adaptive clipping bound learning rate $\eta_C$, threshold multiplier for counting clipped gradients $\tau$, target quantile $\gamma$, the lower-bound of adaptive clipping bound $C_{\text{LB}}$.

**if** unbounded adaptive clipping is used **then**
    Set $C_{\text{LB}} = 0$
**end if**
**for** $t = 0, 1, \ldots, T-1$ **do**
    $\mathcal{B}_t \leftarrow$ Poisson sample a batch with rate $q$ from $D$
    **for** $(x_i, y_i) \in \mathcal{B}_t$ **do**
        $g_i \leftarrow \nabla \mathcal{L}(f_{\theta_t}(x_i), y_i)$
        $\bar{g}_i \leftarrow g_i \cdot \min(\frac{1}{C_t}, \frac{1}{||g_i||})$
    **end for**
    $\tilde{g}_t \leftarrow \frac{1}{B}\left(\sum_{i \in \mathcal{B}_t} \bar{g}_i + \mathcal{N}(0, \sigma_{\text{grad}}^2 \mathbf{I})\right)$
    $\theta_{t+1} \leftarrow \text{OptimizerUpdate}(\theta_t, \tilde{g}_t)$
    $b_t \leftarrow |\{i : ||g_i|| > \tau C_t\}|$
    $\tilde{b}_t \leftarrow \frac{1}{B}(b_t + \mathcal{N}(0, \sigma_{\text{count}}^2))$
    $C_{t+1} \leftarrow \max\left(C_{\text{LB}}, C_t \cdot \exp\left(\eta_C(\tilde{b}_t - \gamma)\right)\right)$
**end for**

---

## 3.2 A KEY LIMITATION OF UNBOUNDED ADAPTIVE CLIPPING

The unbounded adaptive clipping method updates the clipping bound solely based on gradient norm statistics, without enforcing the minimum threshold. While the original theoretical analysis by Andrew et al. (2021) assumes non-changing gradient distribution, as training progresses and gradients from well-optimized samples diminish, the estimated proportion of clipped gradients, $\tilde{b}_t$, often falls below the target quantile $\gamma$, causing the bound $C_t$ to shrink further. This iterative decay suppresses gradients from harder or samples from minority groups, limiting their influence and harming accuracy parity.

To illustrate this problem, consider a toy (non-DP) mean estimation task where the adaptive clipping bound collapses toward zero. In this case, unbounded adaptive clipping fails to recover the true mean, whereas our bounded variant continues to provide a valid estimate (see Figure 1, left).

The target distribution is bimodal with 60% of points around 0 and 40% of points around 1, which implies a true mean $\mu = 0.4$. Using the loss function $\mathcal{L}(\hat{\mu}; x) := \frac{1}{2}\sum_i(x_i - \hat{\mu})^2$, the per-sample gradient is $g_i = x_i - \hat{\mu}_t$. We use $\tau = 1$ and target quantile $\gamma = 0.5$. Early in training both classes contribute sizable gradients, but once $\hat{\mu}_t$ approaches the majority value 0, the majority gradients fall beneath the current bound while the minority (ones) still exceed it. Because $\tilde{b}_t < \gamma$, the unbounded rule keeps shrinking $C_t$ until every minority gradient is clipped to the same tiny magnitude, effectively turning the update into a *majority vote*. The estimate is then driven all the way to 0, as traced by the blue curve in Figure 1. In contrast, our bounded scheme (orange) halts the decay of $C_t$, preserving

the influence of the minority gradients, and converges to approximately the correct mean. We observe the same pattern on the higher-dimensional Fashion-MNIST benchmark in Figure 1 (right), where bounded adaptive clipping consistently yields higher accuracy of the worst-performing class than its unbounded counterpart.

### 3.3 BOUNDED ADAPTIVE CLIPPING: MITIGATING DISPARATE IMPACT

To address the limitations of unbounded adaptive clipping, which often results in excessively small clipping bounds, we propose a bounded adaptive clipping mechanism with a tunable lower bound $C_{\text{LB}}$, as described in the lower-bounded version of Algorithm 2. This mechanism ensures that the clipping bound does not shrink below a specified minimum value, allowing the gradients from the challenging samples to continue contributing to the learning.

Returning to the example in Figure 1, we see that bounded adaptive clipping (orange) effectively prevents the exponential decay of the clipping bound seen in unbounded methods (blue) during later stages of training. By enforcing a lower bound, it avoids excessively suppressing the gradients of minority or confusable classes, ensuring that these gradients contribute efficiently to the accumulated updates. This is particularly critical in later epochs, where the majority of samples become well-optimized, leading to smaller gradients. Without a lower bound, gradients from challenging groups risk being overwhelmed by those of well-optimized samples, halting further optimization for these groups.

### 3.4 PRIVACY OF ADAPTIVE CLIPPING

Achieving DP with adaptive clipping requires accounting for the two accesses to the data for the gradients used in the update as well as the counting query needed for adapting the clipping bound. As both of these are based on the Gaussian mechanism, we can obtain their exact composition using Gaussian DP (Dong et al., 2022).

**Lemma 3.1.** *Let two Gaussian mechanisms have sensitivities $\Delta_1$ and $\Delta_2$ and noise multipliers $\sigma_1$ and $\sigma_2$. Under Gaussian DP, their composition is equivalent to a single Gaussian mechanism with sensitivity $1$ and noise multiplier $\sigma_{\text{eff}} = \frac{1}{\sqrt{\left(\frac{\Delta_1}{\sigma_1}\right)^2 + \left(\frac{\Delta_2}{\sigma_2}\right)^2}}$. In the special case $\Delta_1 = \Delta_2 = 1$, this reduces to $\sigma_{\text{eff}} = \left(\sigma_1^{-2} + \sigma_2^{-2}\right)^{-1/2}$.*

This allows us to evaluate the privacy using standard privacy accountants using the following theorem.

**Theorem 3.2.** *The adaptive clipping algorithm in Algorithm 2 is $(\varepsilon, \delta)$-DP with privacy parameters returned by a privacy accountant using $T = T$, $q = q$, $\sigma = \left(\sigma_{grad}^{-2} + \sigma_{count}^{-2}\right)^{-1/2}$.*

Implementation details are provided in Appendix A.4.

## 4 EXPERIMENTAL RESULTS

We focus on evaluating the performance of our proposed bounded adaptive clipping method under two main settings: i) with optimal hyperparameters, i.e., with the hyperparameter values derived from extensive non-DP-HPO (Section 4.2), and ii) with hyperparameters obtained through DP-HPO, with the privacy cost of HPO also accounted for (Section 4.3).

Under setting (i) with optimal hyperparameters, we want to show that our proposed method outperforms the baselines when each algorithm is optimally tuned for the task. With setting (ii) using hyperparameters resulting from DP-HPO, we demonstrate that our method is more robust to the stochasticity in the hyperparameters compared to the baselines.

### 4.1 METHODOLOGY

**Models**  For image recognition, we use ResNet-18 (He et al., 2016), implemented in Timm (Wightman, 2019), with Batch Normalization replaced by Group Normalization (Wu & He, 2020) as is

standard in DP training (Maaten & Hannun, 2020). We also include a simple two-layer convolutional neural network (CNN); see Appendix A.4 for details. For tabular datasets, we adopt logistic regression.

**Datasets**  We use two image datasets and two tabular datasets in the evaluations:

*Fashion MNIST* (Xiao et al., 2017) contains grayscale images of fashion items from 10 balanced categories. The dataset includes 60,000 training samples and 10,000 test samples. This task involves visually similar classes, which often lead to misclassifications.

*Skewed MNIST* (LeCun & Cortes, 2010; Bagdasaryan et al., 2019) In the skewed MNIST dataset (Bagdasaryan et al., 2019; Xu et al., 2021), class 8 is artificially subsampled to 9% of its original size, leaving approximately 600 samples in the training set, compared to 6,000 samples for the other classes.

*Dutch* (Van der Laan, 2000) and *Adult* (Becker & Kohavi, 1996) are tabular census datasets. In both cases, we treat "gender" as the protected attribute and balance the dataset to ensure equal representation across genders, following the setup of Esipova et al. (2023).

Appendix A.1 provides more details on the dataset configurations, including the class subsampling process for skewed MNIST, the definition of confusable classes in Fashion MNIST, and our data pre-processing methods.

**Metrics**  To evaluate disparate impact, we adopt the notion of **accuracy parity**, which requires similar accuracies across groups.

For image classification tasks, we therefore report two metrics: macro-average accuracy, denoted Macro acc (%), and worst-class accuracy, denoted Worst acc (%).

Unlike standard (micro-)average accuracy dominated by the majority classes, macro-average accuracy gives equal weight to each class, and thus better reflects overall utility under accuracy parity:

$$\text{Macro} = \frac{1}{K} \sum_{k=1}^{K} \frac{\text{TP}_k}{N_k}, \tag{1}$$

where $K$ is the number of classes, and $\text{TP}_k, N_k$ denote the number of true positives and the total number of samples for class $k$, respectively.

Worst-case accuracy measures the performance of the least accurately predicted class. This metric directly aligns with our goal of mitigating disparate impacts by reducing disparities across classes (Zafar et al., 2017).

For tabular datasets with binary prediction tasks, we report accuracy separately for each sensitive class, namely, the Female acc (%), Male acc (%). In addition, we also report the standard group-fairness metric *demographic parity* (Barocas et al., 2023):

$$\text{Demographic Parity} = \frac{\min_a \text{PR}_a}{\max_a \text{PR}_a}, \tag{2}$$

where $\text{PR}$ represents the positivity rate for group $a$. A higher value indicates better demographic parity.

**Grid search**  In both Section 4.2 and Section 4.3, the joint grid search includes two key hyperparameters: the learning rate $\eta$ and the clipping parameter: either the fixed bound $C$ for constant clipping, or the lower bound $C_{\text{LB}}$ for the bounded adaptive schemes. For the unbounded method, we set $C_{\text{LB}} = 0$. Moreover, Macro accuracy is adopted as the objective function. To report performance under optimal hyperparameters, we fix the batch size to a near-optimal value in order to control computational cost.

In contrast, the DP-HPO setting incorporates batch size as an additional tunable parameter. This is enabled by the use of randomized search rather than full grid evaluation, allowing exploration of a higher-dimensional hyperparameter space at reduced cost. Furthermore, our DP-HPO setting explicitly accounts for the privacy cost incurred during hyperparameter search. This ensures that the

final privacy guarantee reflects both model training and hyperparameter selection, adhering to DP principles.

For adaptive methods, we fix other hyperparameters: target quantile $\gamma = 0.5$, multiplier $\tau = 2.5$, and clipping bound learning rate $\eta_C = 0.2$, based on preliminary sensitivity analysis. Appendix A.2 details the full search protocol.

**Baselines** We compare our method against the following baselines: (i) *constant clipping* (Abadi et al., 2016; De et al., 2022) with a constant tuned clipping bound, representing the standard DP-SGD approach; (ii) *unbounded adaptive clipping* (Andrew et al., 2021; Esipova et al., 2023), the current SOTA for mitigating disparate impact in DP-SGD; (iii) *automatic clipping (AUTO)* (Bu et al., 2023), effectively an extreme case of constant clipping that highlights the disparate impact caused by overly small clipping bounds.

We additionally include *FairDP* (Tran et al., 2025), which applies group-specific clipping and noise parameters; results are provided in Appendix C.1. Finally, to reduce the tuning cost of selecting $C_{LB}$, we also evaluate a heuristic variant of our method using a fixed $C_{LB} = 0.1$ across all settings, as detailed in Appendices B.3 and C.2.

## 4.2 RESULTS WITH OPTIMAL HYPERPARAMETERS

We first assess the performance of each clipping strategy under optimal hyperparameter settings. This isolates the capability of each algorithm when tuning is unconstrained.

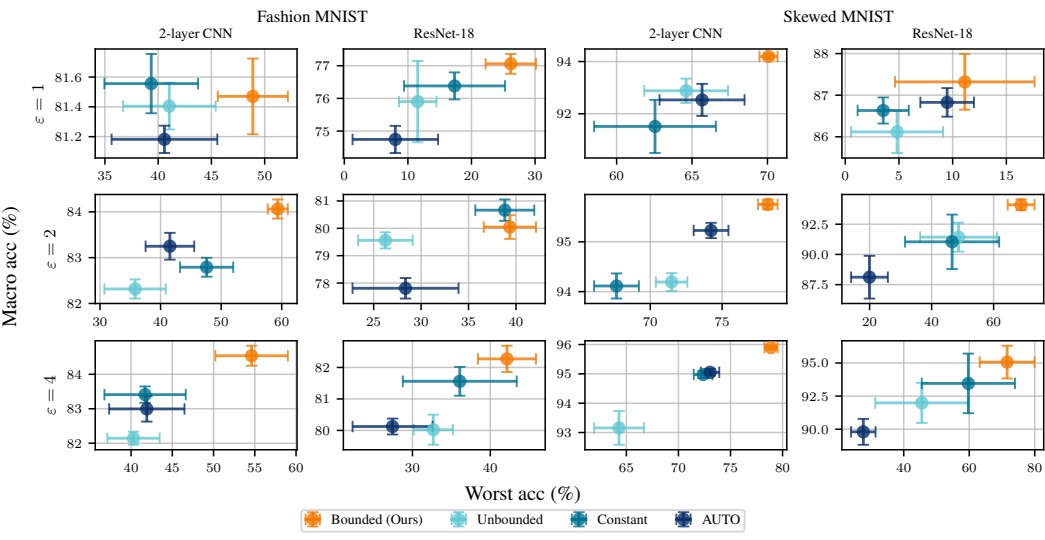

Figure 2: Scatter plots of macro accuracy versus worst-class accuracy across privacy budgets, using optimally tuned hyperparameters. We evaluate four clipping strategies: constant clipping, unbounded adaptive clipping, AutoClip, and our proposed bounded adaptive clipping. Bounded adaptive clipping consistently lies on the empirical Pareto frontier, occupying the upper-right region and achieving strong worst-class accuracy without sacrificing macro accuracy significantly. Each point represents the mean over 10 runs, with standard-error bars ($\delta = 10^{-5}$).

**Image datasets** As shown in Figure 2, bounded adaptive clipping tends to outperform or match alternative strategies across privacy budgets $\varepsilon$, with particularly notable improvements in worst-class accuracy for the two-layer CNN at medium and high privacy budgets, while macro accuracy remains competitive. The results highlight that our algorithm lies on the empirical Pareto frontier, namely, there is no algorithm that can simultaneously achieve higher macro accuracy *and* higher worst-class accuracy. Bounded adaptive clipping, therefore, provides the best available trade-off between overall utility and subgroup robustness.

However, worst-class accuracy does not always increase monotonically with $\varepsilon$, particularly on Fashion MNIST with the two-layer CNN (Figure 2). As the noise level decreases, the optimizer

increasingly prioritizes majority classes, occasionally neglecting minority ones and flattening or even degrading worst-class performance. For training from scratch with ResNet-18, a more challenging setting, bounded adaptive clipping again achieves the best results on Skewed MNIST at medium and high privacy budgets and competitive performance on Fashion MNIST (Figure 2). In this case, AUTO performs substantially worse, suggesting that preserving gradient magnitude information and mitigating misalignment are especially critical in high-dimensional optimization.

These patterns reflect inherent trade-offs in clipping strategies. Constant clipping lacks adaptability and underperforms across both metrics. Unbounded adaptive clipping dynamically follows gradient norms but often drives the bound too low, disproportionately suppressing gradients from minority classes. AUTO rescales all gradients to the sensitivity, effectively setting a clipping bound below most gradient norms, which discards magnitude information and harms the optimization of samples with large gradients. In contrast, our bounded variant prevents this excessive suppression via a lower-bound constraint, achieving a better balance between macro and worst-class accuracy and delivering robust performance across privacy budgets.

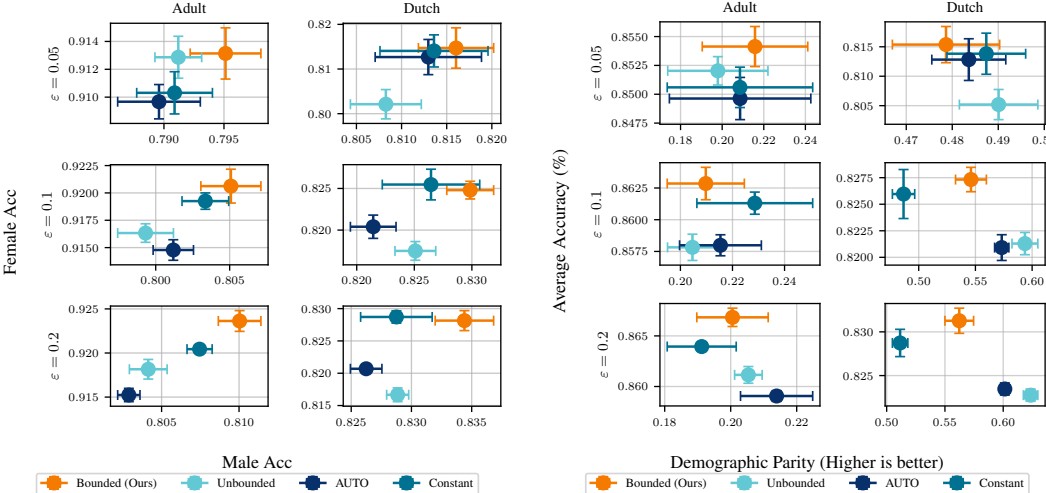

(a) Scatter plots of gender-specific accuracy across privacy budgets $\varepsilon$. Points closer to the upper-right region indicate better simultaneous accuracy for both groups. Bounded adaptive clipping achieves balanced accuracy between genders and remains competitive across all privacy levels.

(b) Scatter plots of average accuracy versus demographic parity across $\varepsilon$. Points nearer the upper-right corner indicate better utility and lower disparate impact. Methods with high demographic parity often show lower accuracy, reflecting that disparate-impact scores can appear improved when utility is degraded.

Figure 3: Scatter plot comparison on the Adult and Dutch tabular datasets using logistic regression trained with differential privacy under four clipping strategies: constant clipping, unbounded adaptive clipping, AutoClip, and bounded adaptive clipping. For each $\varepsilon$, models are tuned for best utility. Across both datasets, bounded adaptive clipping consistently lies on the empirical Pareto frontier, indicating lower disparate impact without sacrificing overall accuracy. Results are averaged over 10 runs with standard-error bars ($\delta = 10^{-5}$).

**Tabular datasets**    Bounded adaptive clipping consistently matches or exceeds the best performance across genders and $\varepsilon$ levels in Figure 3a.

On the Adult dataset, where subgroup accuracies differ markedly, our method improves the performances for both groups. On the Dutch dataset, where subgroup accuracies are more balanced, bounded clipping yields comparable accuracy for females and slightly better accuracy for males than the others.

Unbounded adaptive clipping is consistently weaker, reflecting its tendency to shrink the clipping bound too aggressively and suppress subgroup gradients. AUTO has a similar issue, as it effectively applies a clipping bound smaller than most gradient norms, discarding magnitude information and harming optimization of large-gradient samples. In contrast, bounded adaptive clipping preserves magnitude information and achieves fairer, more stable performance across subgroups.

To further contextualize these results, we examine demographic parity on the tabular datasets in Figure 3b. Across these datasets, bounded adaptive clipping achieves competitive demographic-parity scores while simultaneously attaining high group accuracies. This places our method in the upper-right region of the accuracy–parity plots, showing that the reduced disparate impact is not a consequence of degraded performance but reflects a genuine improvement in group outcomes. The resulting accuracy–parity trade-off exhibits a clear Pareto structure: no alternative method achieves strictly better accuracy without reducing demographic parity, nor strictly better parity without losing accuracy. Thus, bounded adaptive clipping lies on the empirical Pareto frontier.

More broadly, demographic parity and high predictive accuracy are often mutually constraining (Defrance & De Bie, 2025), and our results reflect this theoretical tension. Methods that achieve very high demographic-parity scores typically do so at the cost of substantially lower accuracy, illustrating a limitation of this metric: it can appear "fair" simply because the model is uninformative. In contrast, bounded adaptive clipping maintains strong utility while improving fairness, avoiding misleading fairness gains that arise from uniformly low-utility classifiers.

Together with the image-classification experiments, these results demonstrate the generality of our approach: it not only performs well in complex, high-variance deep learning tasks, but also remains effective in the constrained, low-capacity settings typical of DP learning on tabular data. The analyses further show that bounded adaptive clipping consistently lies on the empirical Pareto frontier across datasets and privacy budgets. Appendix B.3 presents the landscape of the hyperparameter space, confirming the robustness of bounded clipping over a wide range of hyperparameter settings.

Introducing additional hyperparameters increases the cost of hyperparameter optimization. To mitigate this overhead, we propose a simple heuristic (use constant $C_{LB=0.1}$ across settings) that substantially reduces tuning effort while retaining strong performance. The results are described in Appendix C.2, demonstrating close to optimal performance across all settings.

### 4.3 Results with DP-HPO

To assess our adaptive clipping method within a differentially private hyper-parameter optimization (DP-HPO) setting, we begin with a fixed Cartesian grid over the learning rate and clipping-related hyperparameters, then follow Theorem 2 of Papernot & Steinke (2022): the number of grid points we actually evaluate is drawn from a truncated negative-binomial distribution. This randomized stopping rule makes the HPO stage itself DP and allows us to account precisely for the privacy budget consumed during tuning.

We evaluate the full grid to provide a reference with optimal hyperparameters: the total number of grid cells is considered as the expected number of trials in the randomized stopping algorithm, from which the total $\varepsilon$ is computed and marked as the last point in each plot in fig. 4.

**Image datasets** As shown in Figure 4a, AUTO explores a smaller grid since one hyperparameter is fixed, which can help when the number of HPO trials is very limited. Unbounded adaptive clipping also reduces the dimension of grid but performs poorly, as discussed in Section 3.2. In contrast, with moderate to high $\varepsilon$, our bounded adaptive clipping consistently achieves better worst-class accuracy compared to the other baselines.

On Fashion-MNIST, AUTO performs reasonably at small privacy budgets but its best Worst accuracy remains lower than alternatives, showing the negative effect of discarding gradient magnitude. Bounded adaptive clipping, however, provides a clear boost at medium to high budgets, especially for confusable classes.

For skewed MNIST, macro accuracy remains broadly similar across algorithms. Bounded adaptive clipping, however, achieves performance at least on par to the others. At medium to large privacy budgets our bounded adaptive algorithm tends to reach better performance. A zoomed-in view of the moderate-to-high $\varepsilon$ regime for both image datasets is provided in Figure A2.

Overall, DP noise and limited trials hinder HPO from reaching optimality, and the loss of gradient magnitude information further restricts performance. Stable methods like bounded adaptive clipping mitigate these issues, achieving near-optimal worst-class accuracy whenever the number of trials is not extremely limited.

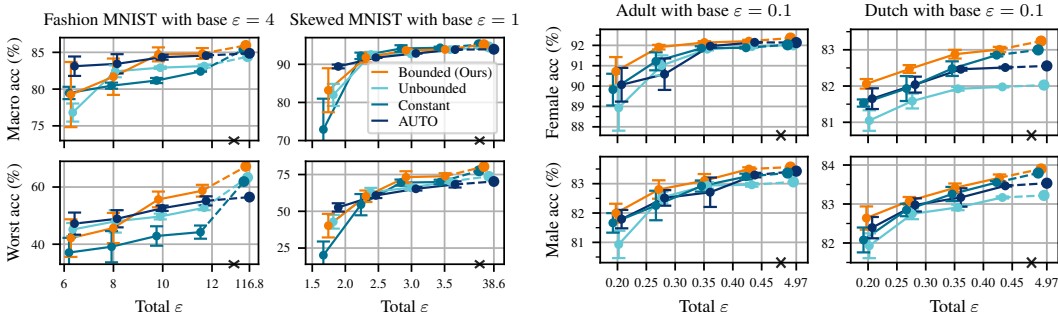

(a) Fashion-MNIST and Skewed MNIST with a two-layer CNN: macro and worst-class accuracy under varying HPO budgets.

(b) Adult and Dutch datasets with logistic regression: gender-specific accuracy under varying HPO budgets.

Figure 4: Performance across privacy budgets ($\varepsilon$) with DP-HPO. The rightmost marker denotes the best accuracy from the full grid search, where the grid size is treated as the expected number of trials in the randomized stopping algorithm. On image datasets, bounded adaptive clipping improves worst-class accuracy on Fashion-MNIST at moderate to high $\varepsilon$ and achieves the best or comparable performance to the others on Skewed MNIST. On tabular datasets, our algorithm matches or exceeds the accuracy of other methods for the lower-performing subgroup, while maintaining comparable performance on the remaining groups. Each curve reports the mean over 10 runs with standard error bars ($\delta = 10^{-5}$); points are jittered for readability.

**Tabular datasets** As shown in Figure 4b, bounded adaptive clipping improves gender-specific performance on the Dutch and Adult datasets, always outperforming or being on par with the best baseline. The gains are most visible at moderate total $\varepsilon$, where bounded clipping often reaches near-optimal accuracy with fewer HPO trials. At very low $\varepsilon$, however, the advantages are less consistent due to noise variability.

## 5 CONCLUSION

This work studies the intersection of DP and disparate impact. While DP ensures that the influence of any individual (or a small group, via group privacy) on the algorithm's output is limited, these guarantees do not address disparities in model accuracy across demographic subgroups. In practice, smaller or minority groups often suffer reduced accuracy under DP training. Group privacy bounds, which degrade exponentially with group size, are too weak to capture this phenomenon and thus cannot explain the disparate-impact issues observed empirically.

To improve the practical disparate-impact behavior of DP learning, this work introduces bounded adaptive clipping, a novel mechanism aimed at mitigating the disparate impacts caused by clipping under differential privacy. By introducing a tunable lower bound for clipping, our method reduces excessive suppression of gradients from minority and confusable groups, alleviating disparate impacts and improving worst-class performance. In particular, our analyses demonstrate that bounded adaptive clipping lies on the empirical Pareto frontier: no alternative method simultaneously attains higher overall utility and lower disparate impact or higher worst-group performance.

Our key findings highlight the advantages of bounded adaptive clipping, including significant improvements in worst-class accuracy and improved robustness, both with optimal hyperparameters and during differentially private hyperparameter tuning. By providing smoother hyperparameter landscapes and achieving competitive performance with small total $\varepsilon$, our approach alleviates the challenges associated with HPO under privacy constraints.

**Limitations** All methods that adapt or tune clipping bounds introduce additional hyperparameters, which expand the search space and reduce the efficiency of DP hyperparameter optimization. This limitation is inherent to the class of approaches, including ours.

## ETHICS STATEMENT

This work does not involve human subjects, personally identifiable data, or sensitive user information. All experiments are conducted on publicly available benchmark datasets. Our focus is on developing methods that mitigate disparate impacts of differentially private training, aiming to improve accuracy parity for underrepresented and confusable groups. While the techniques we propose could be applied broadly, we emphasize responsible use in contexts where fairness, privacy, and equity are critical considerations. We disclose no conflicts of interest or external sponsorship that may have influenced this research.

## REPRODUCIBILITY STATEMENT

We have taken several steps to ensure reproducibility of our results. The Section 4.1 and Appendix A describe all models, algorithms, and hyperparameter settings in detail. Data pre-processing steps and evaluation protocols are clearly documented, and we use publicly available benchmark datasets. Nonetheless, the appendix provides full details on the computing infrastructure, data pre-processing methods, network architectures, training times, and grid search specifications. To facilitate verification, we will release the complete source code and scripts upon acceptance of the paper, ensuring that others can fully replicate our experiments and results.

## LLM USAGE DECLARATION

We used large language models (LLMs) as general-purpose assistive tools. Their use was limited to editing text for grammar, spelling, and word choice; clarifying or summarizing technical concepts; assisting with text formatting; and exploring alternative ideas for data visualization. LLMs did not contribute to research ideation, algorithm design, experimental execution, or result interpretation.

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

## A EXPERIMENTAL DETAILS

### A.1 DATASETS AND PRE-PROCESSSING

In this section, we describe the datasets used in our experiments and the pre-processing steps applied to ensure compatibility with our evaluation objectives.

**Skewed MNIST**

The skewed MNIST dataset is utilized to investigate scenarios where certain groups are underrepresented. Specifically, following prior work (Bagdasaryan et al., 2019; Xu et al., 2021; Esipova et al., 2023), we create an imbalanced training set by sampling only about 10% of the examples from class 8, while and retaining the standard (balanced) test set. The protected feature in this dataset is the class label, which allows us to study disparate impact across different classes.

**Fashion MNIST**

The Fashion MNIST dataset is chosen to study scenarios where some classes are confusable due to feature overlap. Notably, the "Pullover," "Coat," and "Shirt" classes have the highest false positive and false negative rates. This dataset is balanced across classes, making it suitable for evaluating how adaptive clipping mechanisms handle class-specific confusion.

**Adult**

The Adult dataset is used to evaluate disparate impact with respect to sensitive attributes rather than class accuracy. Following the pre-processing steps outlined in (Xu et al., 2021; Esipova et al., 2023), we remove the "final-weight" feature and simplify the "race" attribute to a binary feature (white, non-white). Numerical features are normalized, and categorical features are one-hot encoded. The protected attribute for this dataset is "gender", while the target variable is binary income classification (above or below $50,000).

**Dutch**

The Dutch dataset (Van der Laan, 2000) is used to examine disparate impact in predictions concerning the protected attribute "gender." Pre-processing involves removing underage samples and the "weight" feature, along with filtering out "unemployed" samples and those with missing or middle-level occupation values. Occupation levels are binarized, with codes 4, 5, and 9 classified as low-level professions and codes 1 and 2 as high-level professions. The task is to predict occupation categories based on remaining features.

### A.2 EXPERIMENT ENVIRONMENT AND SETTINGS

Our experiments are performed on the clusters, which equipped with AMD EPYC Trento CPU and AMD MI250x GPU for experiments with image dataset, and Xeon Gold 6230 CPU and Nvidia V100 GPU for tabular datasets.

On image datasets, training a two-layer CNN from scratch takes about 10 minutes on a single GPU, while training a ResNet from scratch requires about 30 minutes. For tabular datasets, about 4 minutes are required to execute one training-from-scratch task.

**The sensitivity of hyperparameters** To assess whether the hyperparameters in the adaptive-clipping mechanism requires precise tuning, we performed a sensitivity study on Adult dataset using Bayesian optimization. All trials shown in Figure A1 were included in this analysis, covering three adaptive-specific hyperparameters: $\tau$ (count threshold), $\gamma$ (target quantile), and $\eta_C$ (learning rate for updating the clipping bound).

The results reveal the pattern: although the Bayesian optimizer explores a wide range of hyperparameter values, broad ranges of $\tau$, $\gamma$, and $\eta_C$ all yield similarly high accuracy. This demonstrates that the adaptive-clipping hyperparameters have low sensitivity with respect to accuracy. The mean and standard deviation of the tuned values are summarized in Table A1, and we therefore use the mean values in our subsequent grid search.

To complement this single-dataset sensitivity view, Table A1 summarizes the mean and standard deviation of the tuned adaptive-specific hyperparameters across four datasets with Bayesian optimiza-

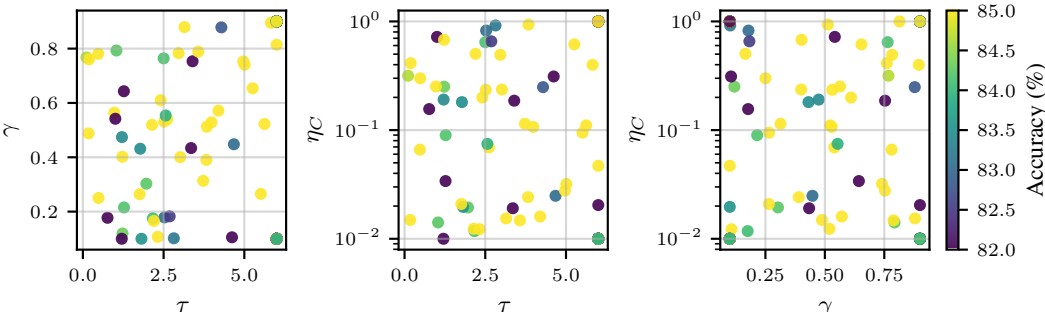

Figure A1: Accuracy stability of adaptive-clipping hyperparameters on Adult using Logistic Regression. Each panel shows pairwise relationships between two hyperparameters $(\tau, \gamma, \eta_C)$, with mean accuracy indicated by color. Although the color gradient appears strong, the absolute accuracy variation is modest. High-performing configurations are distributed broadly across the search space, forming a wide performance plateau. This pattern indicates that adaptive-clipping hyperparameters are insensitive within a large feasible region, supporting our choice to fix them for subsequent experiments.

tion. These statistics further indicate that although the optimizer may assign different absolute values across datasets, the resulting accuracy remains stable, confirming insensitivity to such variations.

Table A1: Mean and standard deviation of adaptive-specific hyperparameters after tuning, across four datasets.

| Hyperparameter | Dataset | Mean | Std |
|---|---|---|---|
| **Batch size** | Skewed MNIST | 15070.6 | 5054.9 |
| | Fashion MNIST | 5992.9 | 1641.3 |
| | Dutch | 8556.6 | 2775.0 |
| | Adult | 12320.0 | 4954.8 |
| **Target quantile ($\gamma$)** | Skewed MNIST | 0.5150 | 0.1691 |
| | Fashion MNIST | 0.4828 | 0.2744 |
| | Dutch | 0.6429 | 0.1713 |
| | Adult | 0.5087 | 0.3267 |
| **Clipping multiplier ($\tau$)** | Skewed MNIST | 2.3002 | 1.6107 |
| | Fashion MNIST | 2.7438 | 3.1248 |
| | Dutch | 2.3864 | 2.0281 |
| | Adult | 2.8830 | 2.0834 |
| **Clipping bound LR ($\eta_C$)** | Skewed MNIST | 0.4250 | 0.3932 |
| | Fashion MNIST | 0.4955 | 0.4067 |
| | Dutch | 0.1402 | 0.1488 |
| | Adult | 0.1691 | 0.1610 |

This empirical robustness supports our decision to treat $C_{\mathrm{LB}}$ as a separate tuning dimension while fixing the remaining hyperparameters in subsequent experiments. Specifically, we fixed $\gamma = 0.5$, $\tau = 2.5$, and $\eta_C = 0.2$ across all datasets, based on their convergence and consistency. The influence of $C_{\mathrm{LB}}$ on training dynamics is further explored in Appendix B.3.

**Seeds** We selected seeds starting from 1. For instance, if 5 seeds were used, the set of seeds would be 1, 2, 3, 4, 5.

**Privacy accounting** To report the $(\varepsilon, \delta)$-DP guarantees, $\delta$ was fixed at $10^{-5}$ across all datasets. For Rènyi Differential Privacy (RDP), we used the hyperparameters outlined in Table A2 and relied on Opacus's implementation (Yousefpour et al., 2021).

**Initial value of clipping bound** The initialization of the clipping bound is a critical aspect of adaptive clipping. Due to the geometric update mechanism introduced by Andrew et al. (2021), the adaptive clipping bound can adjust dynamically across several orders of magnitude during training. This allows the algorithm to efficiently adapt the clipping bound based on the gradient norms observed at each step, accommodating varying distributions of gradient magnitudes. To simplify the process and ensure stability during the initial phases of training, we set the initial clipping bound to 1 across all experiments. This choice strikes a balance between simplicity and generality, as the geometric updates quickly adapt the clipping bound to appropriate levels during training. Our experiments demonstrated that this initialization works effectively across diverse datasets and hyperparameter settings, further highlighting the robustness of the adaptive clipping mechanism. By keeping the initialization consistent, we also reduce the number of hyperparameters requiring fine-tuning, making the method more practical for real-world applications.

Table A2: Hyperparameter values used in experiments for each dataset. The table specifies the Batch Size, Target Quantile ($\gamma$), Threshold Multiplier ($\tau$), and Clipping Bound Learning Rate ($\eta_C$) for each dataset.

| Hyperparameter | Skewed MNIST | Fashion MNIST | Dutch | Adult |
|---|---|---|---|---|
| Epochs | 50 | 50 | 40 | 40 |
| Batch Size | 12,000 | 6,000 | 10,000 | Full-batch |
| Target Quantile ($\gamma$) | 0.5 | 0.5 | 0.5 | 0.5 |
| Threshold Multiplier ($\tau$) | 2.5 | 2.5 | 2.5 | 2.5 |
| Clipping Bound Learning Rate ($\eta_C$) | 0.2 | 0.2 | 0.2 | 0.2 |

### A.2.1 HYPERPARAMETERS USED IN EXPERIMENTS

Based on the evidence and discussion about the sensitivity of some hyperparameters, we used a fixed set of hyperparameters in our experiments to ensure consistency across different datasets. These values include epochs, batch size, target quantile ($\gamma$), threshold multiplier ($\tau$), and clipping bound learning rate ($\eta_C$). Table A2 lists the hyperparameter values used for skewed MNIST, Fashion MNIST, Dutch, and Adult datasets.

It is worth noting that the grid search focused on the learning rate and clipping bound lower bound ($C_{\text{LB}}$). These parameters were excluded from the table as their selection involved a separate evaluation to identify optimal ranges for different datasets. The impact of these parameters is detailed in Appendix B.3.

### A.3 GRID DESIGN

### A.3.1 GRID FOR REPORTING PERFORMANCES WITH OPTIMAL HYPERPARAMETERS

To report the true performances of different algorithms, the experiments should reduce the randomness from hyperparameter optimization. Specifically, all of the reasonable combination of hyperparameters should be tested. Nonetheless, considering the computation cost, we fixed the less sensitive hyperparameters

### A.3.2 GRID FOR RANDOM SEARCH IN DP-HPO

Moreover, random search is widely used to enable privacy accounting in DP-HPO (Liu & Talwar, 2019; Papernot & Steinke, 2022). This approach also requires a predefined grid, from which random samples are drawn.

Table A3: Hyperparameter search space used during tuning. All parameters were treated as categorical.

| Hyperparameter | Ranges |
|---|---|
| Batch size | 1024, 2048, 4096, 8192, 16384, 32768 |
| Learning rate | 1.0000, 1.2915, 1.6681, 2.1544, 2.7826, 3.5938, 4.6416, 5.9948, 7.7426, 10.0000 |
| Learning rate* | 1.0000, 1.2915, 1.6681, 2.1544, 2.7826, 3.5938, 4.6416, 5.9948, 7.7426, 10.0000 20.0000 24.0225 28.8540 34.6572 41.6277 50.0000 |
| Clipping bound | 0.0010, 0.0018, 0.0031, 0.0055, 0.0098, 0.0172, 0.0305, 0.0539, 0.0952, 0.1682, 0.2973, 0.5254, 0.9285, 1.6409, 2.9000, 5.1252, 9.0579, 16.0082, 28.2915, 50.0000 |

\* only for AUTO with tabular datasets.

## A.4 IMPLEMENTATION DETAILS

We build our work on Opacus (Yousefpour et al., 2021), a framework designed for training models with differential privacy. Below, we describe the network architectures used in our experiments.

**Logistic regression**

The logistic regression model consists of a single linear layer that maps the input features directly to the output. The output is passed through a sigmoid activation function to produce probabilities for binary classification tasks.

**Convolutional neural network (CNN)**

The CNN model has the following structure, following (Koskela & Honkela, 2020),

- Two convolutional layers, each followed by a max-pooling layer. The first convolutional layer has 64 filters with a kernel size of 3, followed by a max-pooling layer with a kernel size of 3 and stride 2. The second convolutional layer also has 64 filters with similar configurations.

- Three fully connected layers: the first two layers have 500 units each, and the final fully connected layer outputs predictions for the number of classes in the task.

- ReLU activation functions are used between layers to introduce non-linearity.

**ResNet-18**

We use the ResNet-18 architecture (He et al., 2016) for image classification tasks, implemented via the `timm` library (Wightman, 2019). To comply with standard practices in differentially private training, we replace all Batch Normalization layers with Group Normalization (Wu & He, 2020). All ResNet models are trained from scratch, without any use of pretrained weights.

These architectures are optimized for DP training, ensuring compatibility with privacy constraints while maintaining competitive performance.

## B   FULL RESULT OF EXPERIMENTS

### B.1   DATA SUMMARY

#### B.1.1   RESULTS WITH OPTIMAL HYPERPARAMETERS

The data presented in the Figure 2 are reported in detail as tables in Tables A4 and A5. Moreover, the Figure 3 is detailed in Table A6.

The *Fix-Bounded Adaptive* variant corresponds to our *Bounded Adaptive* algorithm using a fixed lower bound of $C_{LB} = 0.1$. It is designed to reduce the computational cost associated with tuning the optimal value of $C_{LB}$.

Table A4: Comparison of macro accuracy and worst-class accuracy across algorithms on the Fashion MNIST and Skewed MNIST datasets under varying privacy budgets ($\varepsilon$) with ResNet-50. Bounded adaptive clipping consistently achieves higher worst-class accuracy while maintaining competitive macro accuracy.

| Dataset | $\varepsilon$ | Algorithm | Macro Acc. | Worst-Class Acc. |
|---|---|---|---|---|
| Fashion MNIST | 1.0 | Bounded Adaptive (Ours) | $0.7706 \pm 0.0030$ | $\mathbf{0.2618} \pm 0.0397$ |
| | | Fix-Bounded Adaptive (Ours) | $0.7660 \pm 0.0055$ | $\mathbf{0.2539} \pm 0.0265$ |
| | | Constant Clipping | $0.7638 \pm 0.0041$ | $0.1734 \pm 0.0794$ |
| | | Unbounded Adaptive | $0.7590 \pm 0.0125$ | $0.1154 \pm 0.0296$ |
| | | AUTO | $0.7474 \pm 0.0042$ | $0.0802 \pm 0.0673$ |
| | 2.0 | Bounded Adaptive (Ours) | $0.8005 \pm 0.0043$ | $0.3936 \pm 0.0275$ |
| | | Fix-Bounded Adaptive (Ours) | $0.7897 \pm 0.0048$ | $0.3503 \pm 0.0297$ |
| | | Constant Clipping | $0.8066 \pm 0.0039$ | $0.3882 \pm 0.0311$ |
| | | Unbounded Adaptive | $0.7956 \pm 0.0030$ | $0.2624 \pm 0.0287$ |
| | | AUTO | $0.7781 \pm 0.0038$ | $0.2836 \pm 0.0558$ |
| | 4.0 | Bounded Adaptive (Ours) | $0.8227 \pm 0.0042$ | $\mathbf{0.4214} \pm 0.0373$ |
| | | Fix-Bounded Adaptive (Ours) | $0.8104 \pm 0.0047$ | $\mathbf{0.4117} \pm 0.0255$ |
| | | Constant Clipping | $0.8156 \pm 0.0046$ | $0.3608 \pm 0.0732$ |
| | | Unbounded Adaptive | $0.8002 \pm 0.0048$ | $0.3268 \pm 0.0252$ |
| | | AUTO | $0.8012 \pm 0.0025$ | $0.2744 \pm 0.0514$ |
| Skewed MNIST | 1.0 | Bounded Adaptive (Ours) | $0.8732 \pm 0.0067$ | $0.1113 \pm 0.0749$ |
| | | Fix-Bounded Adaptive (Ours) | $0.8587 \pm 0.0057$ | $0.1062 \pm 0.0384$ |
| | | Constant Clipping | $0.8663 \pm 0.0031$ | $0.0353 \pm 0.0238$ |
| | | Unbounded Adaptive | $0.8612 \pm 0.0051$ | $0.0483 \pm 0.0428$ |
| | | AUTO | $0.8683 \pm 0.0034$ | $0.0949 \pm 0.0249$ |
| | 2.0 | Bounded Adaptive (Ours) | $\mathbf{0.9411} \pm 0.0044$ | $\mathbf{0.6887} \pm 0.0433$ |
| | | Fix-Bounded Adaptive (Ours) | $0.9223 \pm 0.0076$ | $0.6336 \pm 0.0355$ |
| | | Constant Clipping | $0.9104 \pm 0.0225$ | $0.4661 \pm 0.1518$ |
| | | Unbounded Adaptive | $0.9143 \pm 0.0120$ | $0.4869 \pm 0.1237$ |
| | | AUTO | $0.8812 \pm 0.0177$ | $0.1996 \pm 0.0593$ |
| | 4.0 | Bounded Adaptive (Ours) | $0.9507 \pm 0.0124$ | $0.7157 \pm 0.0837$ |
| | | Fix-Bounded Adaptive (Ours) | $0.9317 \pm 0.0090$ | $0.6441 \pm 0.0754$ |
| | | Constant Clipping | $0.9346 \pm 0.0225$ | $0.5973 \pm 0.1423$ |
| | | Unbounded Adaptive | $0.9199 \pm 0.0151$ | $0.4552 \pm 0.1420$ |
| | | AUTO | $0.8980 \pm 0.0098$ | $0.2766 \pm 0.0374$ |

Table A5: Comparison of macro accuracy and worst-class accuracy across algorithms on the Fashion MNIST and Skewed MNIST datasets under varying privacy budgets ($\varepsilon$) with two-layer CNN. Bounded adaptive clipping consistently achieves higher worst-class accuracy while maintaining competitive macro accuracy.

| Dataset | $\varepsilon$ | Algorithm | Macro Acc. | Worst-Class Acc. |
|---|---|---|---|---|
| Fashion MNIST | 1.0 | Bounded Adaptive (Ours) | $0.8147 \pm 0.0025$ | $\mathbf{0.4889} \pm 0.0329$ |
| | | Fix-Bounded (Ours) | $0.8129 \pm 0.0038$ | $0.4350 \pm 0.0465$ |
| | | Constant Clipping | $0.8156 \pm 0.0020$ | $0.3935 \pm 0.0440$ |
| | | Unbounded Adaptive | $0.8140 \pm 0.0016$ | $0.4105 \pm 0.0435$ |
| | | AUTO | $0.8118 \pm 0.0009$ | $0.4059 \pm 0.0496$ |
| | 2.0 | Bounded Adaptive (Ours) | $\mathbf{0.8406} \pm 0.0021$ | $\mathbf{0.5935} \pm 0.0167$ |
| | | Fix-Bounded (Ours) | $0.8337 \pm 0.0043$ | $0.5128 \pm 0.0223$ |
| | | Constant Clipping | $0.8279 \pm 0.0021$ | $0.4761 \pm 0.0439$ |
| | | Unbounded Adaptive | $0.8232 \pm 0.0021$ | $0.3580 \pm 0.0507$ |
| | | AUTO | $0.8325 \pm 0.0029$ | $0.4154 \pm 0.0402$ |
| | 4.0 | Bounded Adaptive (Ours) | $\mathbf{0.8454} \pm 0.0029$ | $\mathbf{0.5461} \pm 0.0439$ |
| | | Fix-Bounded (Ours) | $0.8389 \pm 0.0025$ | $0.5041 \pm 0.0362$ |
| | | Constant Clipping | $0.8341 \pm 0.0024$ | $0.4170 \pm 0.0493$ |
| | | Unbounded Adaptive | $0.8214 \pm 0.0019$ | $0.4030 \pm 0.0317$ |
| | | AUTO | $0.8300 \pm 0.0037$ | $0.4191 \pm 0.0456$ |
| Skewed MNIST | 1.0 | Bounded Adaptive (Ours) | $\mathbf{0.9419} \pm 0.0010$ | $\mathbf{0.7007} \pm 0.0060$ |
| | | Fix-Bounded (Ours) | $0.9411 \pm 0.0014$ | $0.6830 \pm 0.0097$ |
| | | Constant Clipping | $0.9152 \pm 0.0102$ | $0.6255 \pm 0.0404$ |
| | | Unbounded Adaptive | $0.9288 \pm 0.0047$ | $0.6461 \pm 0.0277$ |
| | | AUTO | $0.9253 \pm 0.0061$ | $0.6566 \pm 0.0282$ |
| | 2.0 | Bounded Adaptive (Ours) | $\mathbf{0.9575} \pm 0.0010$ | $\mathbf{0.7818} \pm 0.0068$ |
| | | Fix-Bounded (Ours) | $0.9507 \pm 0.0017$ | $0.7528 \pm 0.0092$ |
| | | Constant Clipping | $0.9411 \pm 0.0025$ | $0.6765 \pm 0.0156$ |
| | | Unbounded Adaptive | $0.9419 \pm 0.0018$ | $0.7148 \pm 0.0109$ |
| | | AUTO | $0.9523 \pm 0.0015$ | $0.7423 \pm 0.0121$ |
| | 4.0 | Bounded Adaptive (Ours) | $\mathbf{0.9589} \pm 0.0006$ | $\mathbf{0.7889} \pm 0.0063$ |
| | | Fix-Bounded (Ours) | $0.9565 \pm 0.0012$ | $0.7705 \pm 0.0077$ |
| | | Constant Clipping | $0.9498 \pm 0.0012$ | $0.7237 \pm 0.0090$ |
| | | Unbounded Adaptive | $0.9316 \pm 0.0058$ | $0.6429 \pm 0.0239$ |
| | | AUTO | $0.9505 \pm 0.0011$ | $0.7304 \pm 0.0088$ |

Table A6: Accuracy for female and male groups across tabular datasets, algorithms, and privacy budgets ($\varepsilon$). As the differences between our bounded adaptive clipping and the constant clipping baseline are not statistically significant, no values are highlighted.

| Dataset | $\varepsilon$ | Algorithm | Female Acc. | Male Acc. |
|---|---|---|---|---|
| Adult | 0.05 | Bounded Adaptive (Ours) | $0.9131 \pm 0.0018$ | $0.7951 \pm 0.0029$ |
| | | Fix-Bounded Adaptive (Ours) | $0.9131 \pm 0.0018$ | $0.7951 \pm 0.0029$ |
| | | Constant Clipping | $0.9103 \pm 0.0015$ | $0.7909 \pm 0.0032$ |
| | | Unbounded Adaptive | $0.9129 \pm 0.0015$ | $0.7912 \pm 0.0019$ |
| | | AUTO | $0.9097 \pm 0.0012$ | $0.7896 \pm 0.0034$ |
| | 0.10 | Bounded Adaptive (Ours) | $0.9206 \pm 0.0015$ | $0.8051 \pm 0.0020$ |
| | | Fix-Bounded Adaptive (Ours) | $0.9191 \pm 0.0012$ | $0.8044 \pm 0.0017$ |
| | | Constant Clipping | $0.9193 \pm 0.0007$ | $0.8034 \pm 0.0016$ |
| | | Unbounded Adaptive | $0.9163 \pm 0.0008$ | $0.7993 \pm 0.0019$ |
| | | AUTO | $0.9148 \pm 0.0009$ | $0.8012 \pm 0.0014$ |
| | 0.20 | Bounded Adaptive (Ours) | $0.9236 \pm 0.0012$ | $0.8100 \pm 0.0014$ |
| | | Fix-Bounded Adaptive (Ours) | $0.9214 \pm 0.0012$ | $0.8079 \pm 0.0011$ |
| | | Constant Clipping | $0.9204 \pm 0.0004$ | $0.8075 \pm 0.0008$ |
| | | Unbounded Adaptive | $0.9182 \pm 0.0011$ | $0.8041 \pm 0.0012$ |
| | | AUTO | $0.9152 \pm 0.0008$ | $0.8029 \pm 0.0007$ |
| Dutch | 0.05 | Bounded Adaptive (Ours) | $0.8147 \pm 0.0045$ | $0.8160 \pm 0.0042$ |
| | | Fix-Bounded Adaptive (Ours) | $0.8128 \pm 0.0028$ | $0.8088 \pm 0.0045$ |
| | | Constant Clipping | $0.8141 \pm 0.0036$ | $0.8136 \pm 0.0060$ |
| | | Unbounded Adaptive | $0.8021 \pm 0.0033$ | $0.8083 \pm 0.0039$ |
| | | AUTO | $0.8127 \pm 0.0039$ | $0.8130 \pm 0.0059$ |
| | 0.10 | Bounded Adaptive (Ours) | $0.8248 \pm 0.0011$ | $0.8299 \pm 0.0020$ |
| | | Fix-Bounded Adaptive (Ours) | $0.8230 \pm 0.0020$ | $0.8277 \pm 0.0022$ |
| | | Constant Clipping | $0.8255 \pm 0.0018$ | $0.8265 \pm 0.0042$ |
| | | Unbounded Adaptive | $0.8175 \pm 0.0011$ | $0.8251 \pm 0.0018$ |
| | | AUTO | $0.8204 \pm 0.0014$ | $0.8214 \pm 0.0020$ |
| | 0.20 | Bounded Adaptive (Ours) | $0.8281 \pm 0.0015$ | $0.8344 \pm 0.0024$ |
| | | Fix-Bounded Adaptive (Ours) | $0.8258 \pm 0.0019$ | $0.8312 \pm 0.0015$ |
| | | Constant Clipping | $0.8287 \pm 0.0010$ | $0.8288 \pm 0.0030$ |
| | | Unbounded Adaptive | $0.8167 \pm 0.0011$ | $0.8289 \pm 0.0009$ |
| | | AUTO | $0.8207 \pm 0.0005$ | $0.8262 \pm 0.0013$ |

### B.1.2 RESULTS WITH DP-HPO

Table A7: Accuracy for Macro accuracy and Worst-class accuracy across two image datasets, algorithms, and privacy budgets ($\varepsilon$) with DP-HPO.

| Dataset | $\varepsilon$ | Algorithm | Macro Acc. | Worst-Class Acc. |
|---|---|---|---|---|
| Fashion MNIST | 6.273 | Bounded Adaptive | $0.7925 \pm 0.0443$ | $0.4213 \pm 0.0659$ |
| | | Constant Clipping | $0.7947 \pm 0.0084$ | $0.3708 \pm 0.0514$ |
| | | Unbounded Clipping | $0.7681 \pm 0.0123$ | $0.4514 \pm 0.0227$ |
| | | AUTO | $0.8311 \pm 0.0132$ | $0.4720 \pm 0.0387$ |
| | 7.998 | Bounded Adaptive | $0.8168 \pm 0.0247$ | $0.4567 \pm 0.0527$ |
| | | Constant Clipping | $0.8046 \pm 0.0041$ | $0.3912 \pm 0.0546$ |
| | | Unbounded Adaptive | $0.8241 \pm 0.0080$ | $0.4799 \pm 0.0393$ |
| | | AUTO | $0.8344 \pm 0.0131$ | $0.4889 \pm 0.0300$ |
| | 9.825 | Bounded Adaptive | $0.8472 \pm 0.0095$ | $0.5566 \pm 0.0275$ |
| | | Constant Clipping | $0.8117 \pm 0.0035$ | $0.4287 \pm 0.0340$ |
| | | Unbounded Adaptive | $0.8291 \pm 0.0016$ | $0.4970 \pm 0.0101$ |
| | | AUTO | $0.8433 \pm 0.0033$ | $0.5258 \pm 0.0093$ |
| | 11.608 | Bounded Adaptive | $0.8484 \pm 0.0074$ | $0.5871 \pm 0.0206$ |
| | | Constant Clipping | $0.8239 \pm 0.0001$ | $0.4423 \pm 0.0231$ |
| | | Unbounded Adaptive | $0.8313 \pm 0.0021$ | $0.5275 \pm 0.0085$ |
| | | AUTO | $0.8456 \pm 0.0023$ | $0.5516 \pm 0.0159$ |
| | 13.370* | Bounded Adaptive | $0.8594$ | $0.6720$ |
| | | Constant Clipping | $0.8539$ | $0.6190$ |
| | | Unbounded Adaptive | $0.8441$ | $0.6335$ |
| | | AUTO | $0.8487$ | $0.5648$ |
| MNIST | 1.743 | Bounded Adaptive | $0.8320 \pm 0.0577$ | $0.4023 \pm 0.0802$ |
| | | Constant Clipping | $0.7290 \pm 0.0808$ | $0.2011 \pm 0.0934$ |
| | | Unbounded Adaptive | $0.8197 \pm 0.0290$ | $0.4287 \pm 0.0466$ |
| | | AUTO | $0.8943 \pm 0.0049$ | $0.5262 \pm 0.0285$ |
| | 2.311 | Bounded Adaptive | $0.9177 \pm 0.0127$ | $0.6023 \pm 0.0391$ |
| | | Constant Clipping | $0.9223 \pm 0.0084$ | $0.5450 \pm 0.0723$ |
| | | Unbounded Adaptive | $0.9256 \pm 0.0071$ | $0.6295 \pm 0.0298$ |
| | | AUTO | $0.9174 \pm 0.0007$ | $0.6075 \pm 0.0175$ |
| | 2.916 | Bounded Adaptive | $0.9366 \pm 0.0044$ | $0.7320 \pm 0.0404$ |
| | | Constant Clipping | $0.9415 \pm 0.0086$ | $0.6950 \pm 0.0208$ |
| | | Unbounded Adaptive | $0.9272 \pm 0.0023$ | $0.6614 \pm 0.0164$ |
| | | AUTO | $0.9286 \pm 0.0004$ | $0.6527 \pm 0.0114$ |
| | 3.509 | Bounded Adaptive | $0.9388 \pm 0.0038$ | $0.7388 \pm 0.0201$ |
| | | Constant Clipping | $0.9435 \pm 0.0028$ | $0.6990 \pm 0.0156$ |
| | | Unbounded Adaptive | $0.9387 \pm 0.0014$ | $0.7030 \pm 0.0095$ |
| | | AUTO | $0.9384 \pm 0.0004$ | $0.6827 \pm 0.0209$ |
| | 4.095* | Bounded Adaptive | $0.9516$ | $0.8039$ |
| | | Constant Clipping | $0.9515$ | $0.7754$ |
| | | Unbounded Adaptive | $0.9456$ | $0.7377$ |
| | | AUTO | $0.9395$ | $0.7027$ |

The star (*) represents the best result obtained with the optimal hyperparameters; namely, given the number of trials equal to the size of the grid search, this is the best performance that the model can obtain.

Table A8: Accuracy for female and male groups across two tabular datasets, algorithms, and privacy budgets ($\varepsilon$) with DP-HPO.

| Dataset | Total$\varepsilon$ | Algorithm | Female Acc. | Male Acc. |
|---|---|---|---|---|
| Adult | 0.1971 | Bounded Adaptive | $0.9072 \pm 0.0070$ | $0.8199 \pm 0.0032$ |
| | | Constant Clipping | $0.8983 \pm 0.0078$ | $0.8166 \pm 0.0033$ |
| | | Unbounded Adaptive | $0.8893 \pm 0.0112$ | $0.8093 \pm 0.0047$ |
| | | AUTO | $0.9007 \pm 0.0083$ | $0.8179 \pm 0.0031$ |
| | 0.2716 | Bounded Adaptive | $0.9191 \pm 0.0011$ | $0.8280 \pm 0.0031$ |
| | | Constant Clipping | $0.9122 \pm 0.0044$ | $0.8226 \pm 0.0050$ |
| | | Unbounded Adaptive | $0.9101 \pm 0.0049$ | $0.8253 \pm 0.0020$ |
| | | AUTO | $0.9058 \pm 0.0077$ | $0.8251 \pm 0.0027$ |
| | 0.3507 | Bounded Adaptive | $0.9214 \pm 0.0007$ | $0.8310 \pm 0.0023$ |
| | | Constant Clipping | $0.9186 \pm 0.0010$ | $0.8292 \pm 0.0019$ |
| | | Unbounded Adaptive | $0.9181 \pm 0.0006$ | $0.8294 \pm 0.0011$ |
| | | AUTO | $0.9196 \pm 0.0007$ | $0.8271 \pm 0.0050$ |
| | 0.4279 | Bounded Adaptive | $0.9221 \pm 0.0007$ | $0.8350 \pm 0.0006$ |
| | | Constant Clipping | $0.9188 \pm 0.0004$ | $0.8324 \pm 0.0010$ |
| | | Unbounded Adaptive | $0.9194 \pm 0.0006$ | $0.8296 \pm 0.0006$ |
| | | AUTO | $0.9215 \pm 0.0002$ | $0.8329 \pm 0.0008$ |
| | 4.9714* | Bounded Adaptive | 0.9236 | 0.8357 |
| | | Constant Clipping | 0.9203 | 0.8336 |
| | | Unbounded Adaptive | 0.9203 | 0.8306 |
| | | AUTO | 0.9214 | 0.8343 |
| Dutch | 0.1971 | Bounded Adaptive | $0.8206 \pm 0.0013$ | $0.8264 \pm 0.0030$ |
| | | Constant Clipping | $0.8153 \pm 0.0010$ | $0.8208 \pm 0.0032$ |
| | | Unbounded Adaptive | $0.8105 \pm 0.0028$ | $0.8193 \pm 0.0032$ |
| | | AUTO | $0.8165 \pm 0.0029$ | $0.8239 \pm 0.0027$ |
| | 0.2717 | Bounded Adaptive | $0.8247 \pm 0.0011$ | $0.8308 \pm 0.0010$ |
| | | Constant Clipping | $0.8193 \pm 0.0034$ | $0.8285 \pm 0.0016$ |
| | | Unbounded Adaptive | $0.8159 \pm 0.0020$ | $0.8275 \pm 0.0014$ |
| | | AUTO | $0.8204 \pm 0.0022$ | $0.8298 \pm 0.0017$ |
| | 0.3508 | Bounded Adaptive | $0.8288 \pm 0.0012$ | $0.8342 \pm 0.0011$ |
| | | Constant Clipping | $0.8248 \pm 0.0020$ | $0.8326 \pm 0.0018$ |
| | | Unbounded Adaptive | $0.8192 \pm 0.0005$ | $0.8291 \pm 0.0007$ |
| | | AUTO | $0.8246 \pm 0.0004$ | $0.8317 \pm 0.0024$ |
| | 0.4280 | Bounded Adaptive | $0.8301 \pm 0.0007$ | $0.8367 \pm 0.0008$ |
| | | Constant Clipping | $0.8285 \pm 0.0004$ | $0.8355 \pm 0.0004$ |
| | | Unbounded Adaptive | $0.8198 \pm 0.0002$ | $0.8317 \pm 0.0003$ |
| | | AUTO | $0.8251 \pm 0.0002$ | $0.8347 \pm 0.0002$ |
| | 4.9712* | Bounded Adaptive | 0.8324 | 0.8390 |
| | | Constant Clipping | 0.8300 | 0.8380 |
| | | Unbounded Adaptive | 0.8203 | 0.8322 |
| | | AUTO | 0.8255 | 0.8353 |

The star (*) represents the best result obtained with the optimal hyperparameters; namely, given the number of trials equal to the size of the grid search, this is the best performance that the model can obtain.

## B.2 ZOOMED-IN FIGURES

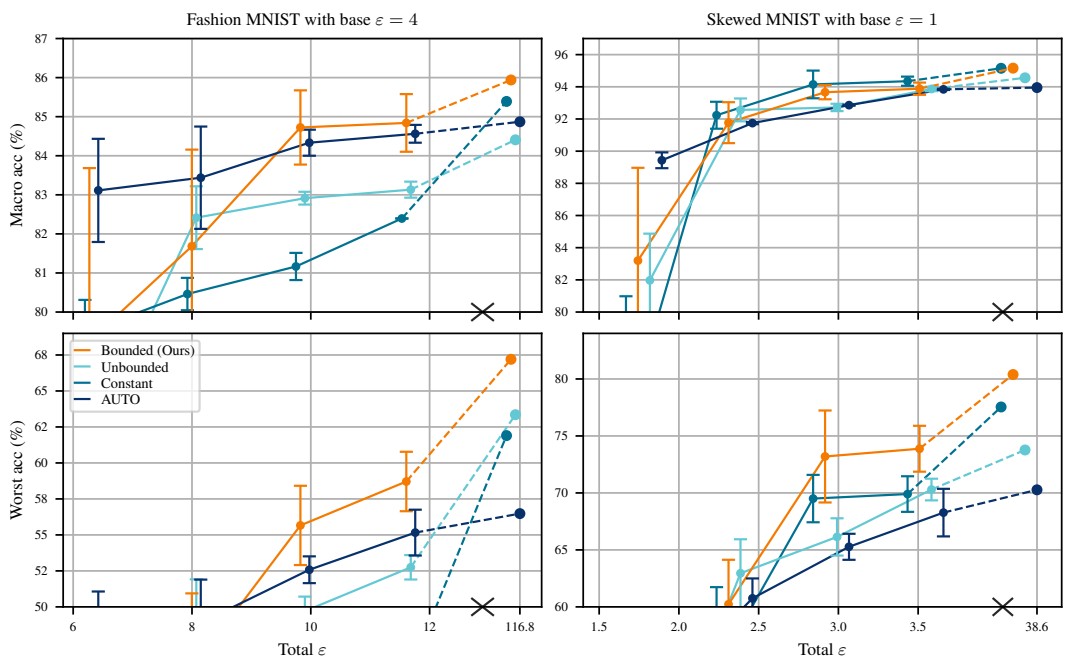

Figure A2: Fashion-MNIST and Skewed MNIST with a two-layer CNN: macro and worst-class accuracy under varying HPO budgets. Zoomed-in version of Figure 4a, highlighting the differences at medium to large privacy budgets.

## B.3 HEAT-MAP OF THE LANDSCAPE AMONG DIFFERENT METRICS ON DATASETS

In this subsection, we provide heatmaps to visualize the landscape of hyperparameter optimization across different metrics for the datasets used in our experiments. The heatmaps illustrate how the learning rate and the clipping bound lower bound ($C_{\mathrm{LB}}$) interact to influence performance across various metrics. Each dataset is analyzed under its specific privacy budget ($\varepsilon$), and the metrics reported are tailored to the characteristics of the dataset.

For Fashion MNIST, a balanced dataset, we report macro accuracy, worst-class accuracy, and loss, omitting micro accuracy as it is nearly identical to macro accuracy, which are shown in Figure A3. The heatmap shows that the hyperparameter landscape for macro and worst-class accuracy largely overlaps, indicating robust performance across different objectives. The landscape of AUTO is shown in Figure A4.

For skewed MNIST, a class-imbalanced dataset, we report macro accuracy, worst-class accuracy, micro accuracy, and loss in Figure A5. The inclusion of micro accuracy highlights the discrepancies between class-weighted metrics (macro) and sample-weighted metrics (micro), showcasing how imbalance affects the optimization landscape. The heatmap reveals that the optimal regions for macro and micro accuracy are closely aligned, but worst-class accuracy demonstrates a more restrictive optimal range, indicating its sensitivity to hyperparameters. The landscape of AUTO is shown in Figure A6.

For Adult and Dutch datasets, we focus on the accuracy gap between genders and the overall loss in Figures A7 and A9. These datasets are used to evaluate fairness-related metrics, with male and female accuracies reported separately. The heatmaps highlight how hyperparameters influence gender disparities in accuracy. While minimizing loss generally aligns with optimizing male and female accuracies, the gender gap exhibits a more nuanced response, requiring careful hyperparameter tuning to ensure utility and fairness. The landscape of AUTO is shown in Figures A8 and A10.

Based on the observation from this chapter, we recommend using a universal value such as $C_{LB} = 0.1$, when computational resources are limited. This value allows the adaptive rule to track early-training dynamics while still preventing late-stage over-clipping, as discussed in Appendix C.2.

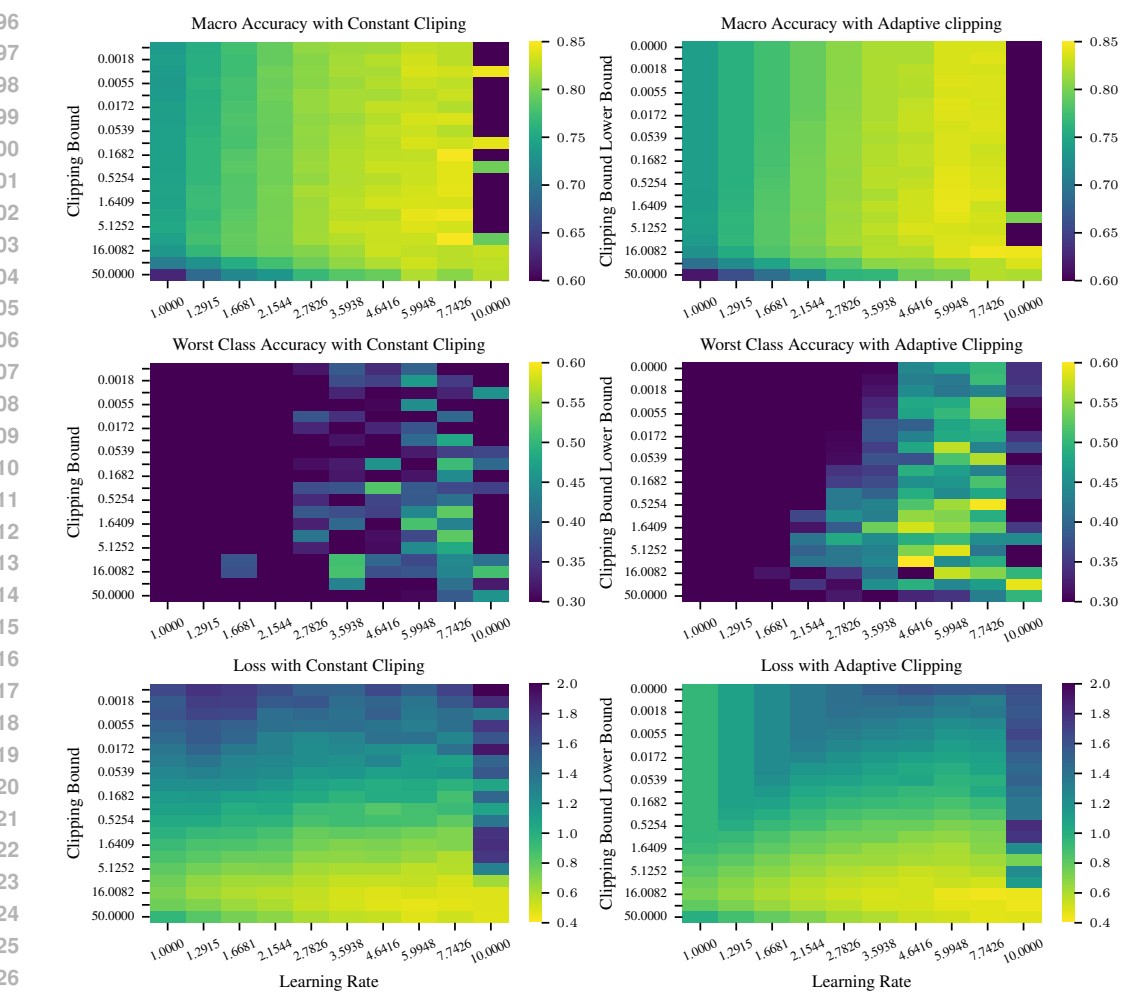

Figure A3: Heatmaps of macro accuracy, worst-class accuracy, and loss on Fashion-MNIST at $\varepsilon = 4.0$ using constant, bounded adaptive, and unbounded adaptive algorithms. The rows correspond to adaptive clipping with different lower bounds $C_{LB}$; the case $C_{LB} = 0$ (first row) represents unbounded adaptive clipping.

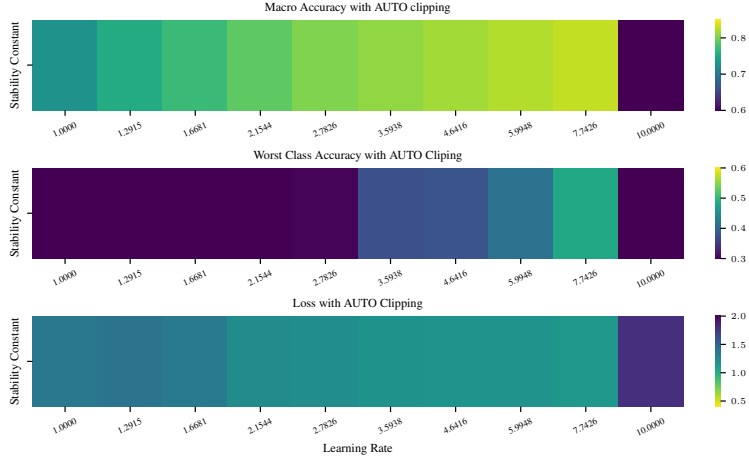

Figure A4: Heatmaps of macro accuracy, worst-class accuracy, and loss on Fashion-MNIST with $\varepsilon = 4.0$ using AUTO. The stability constant is set to the recommended value of $0.01$ Bu et al. (2023).

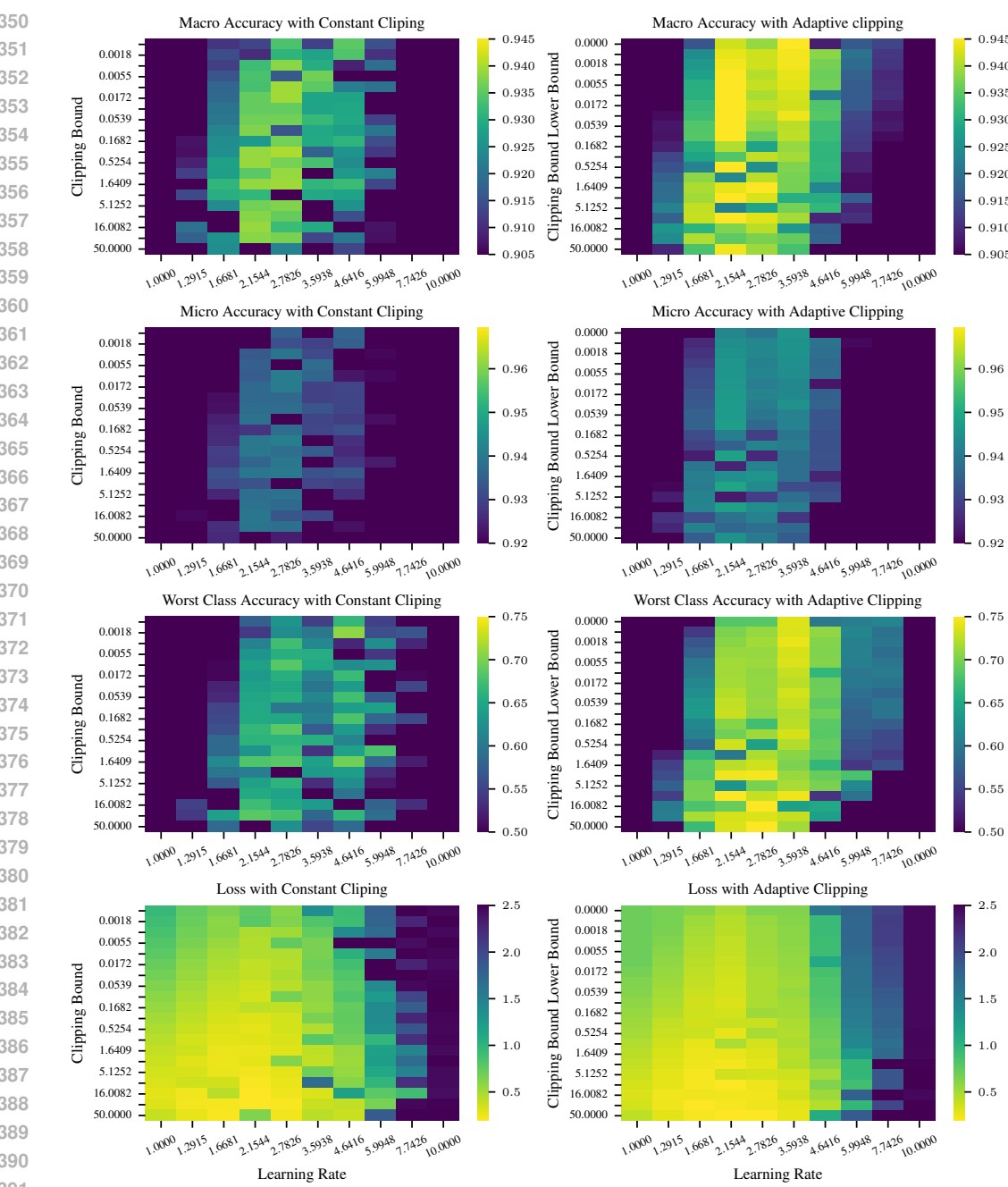

Figure A5: Heatmaps of macro accuracy, micro accuracy, worst-class accuracy, and loss on Skewed MNIST at $\varepsilon = 1.0$ using constant, bounded adaptive, and unbounded adaptive algorithms. The heatmap rows correspond to adaptive clipping with different lower bounds $C_{LB}$; the case $C_{LB} = 0$ (first row) represents unbounded adaptive clipping.

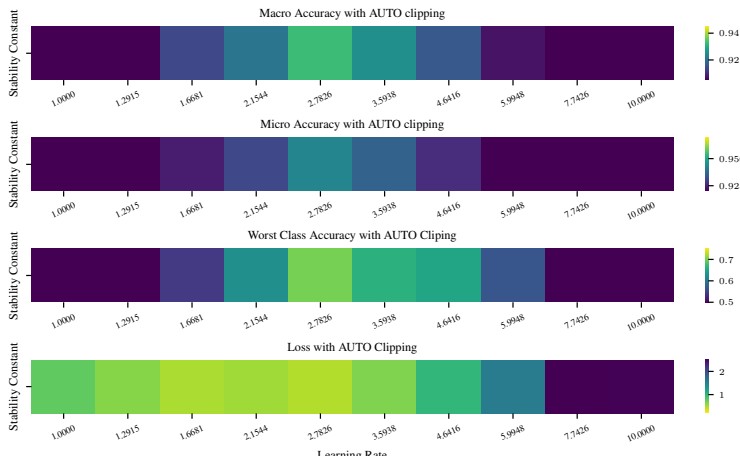

Figure A6: Heatmaps of macro accuracy, micro accuracy, worst-class accuracy, and loss on Skewed MNIST with $\varepsilon = 1.0$ using AUTO. The stability constant is set to the recommended value of $0.01$ Bu et al. (2023).

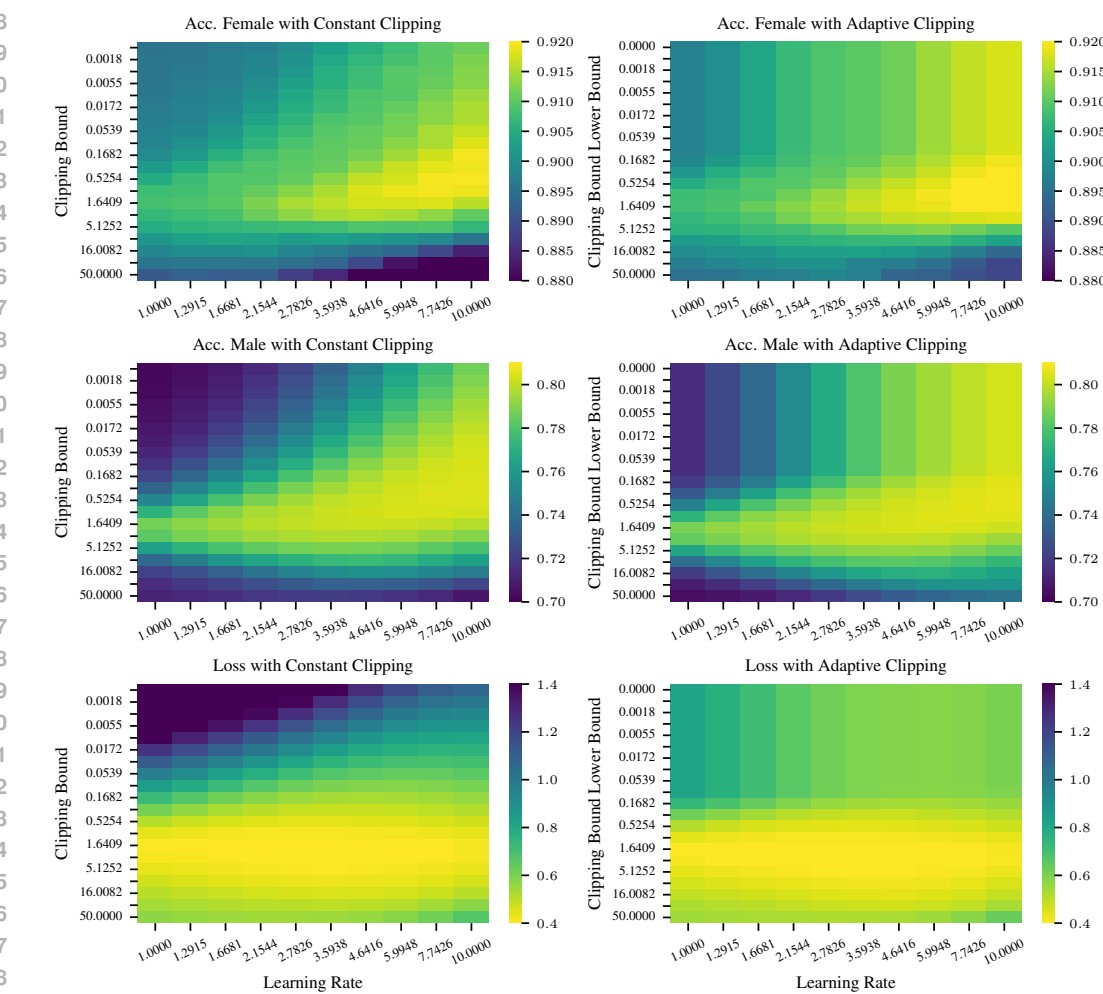

Figure A7: Heatmaps of female accuracy, male accuracy, and loss on Adult at $\varepsilon = 0.1$ using constant, bounded adaptive, and unbounded adaptive algorithms. The heatmap rows correspond to adaptive clipping with different lower bounds $C_{LB}$; the case $C_{LB} = 0$ (first row) represents unbounded adaptive clipping.

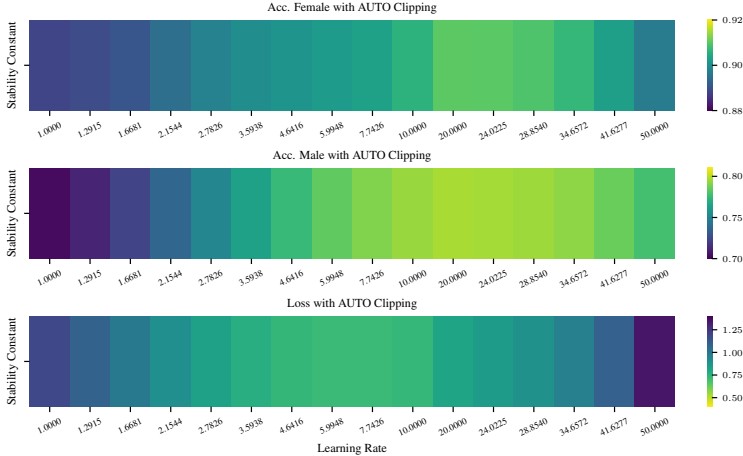

Figure A8: Heatmaps of female accuracy, male accuracy, and loss on Adult with $\varepsilon = 0.1$ using AUTO. The stability constant is set to the recommended value of $0.01$ Bu et al. (2023).

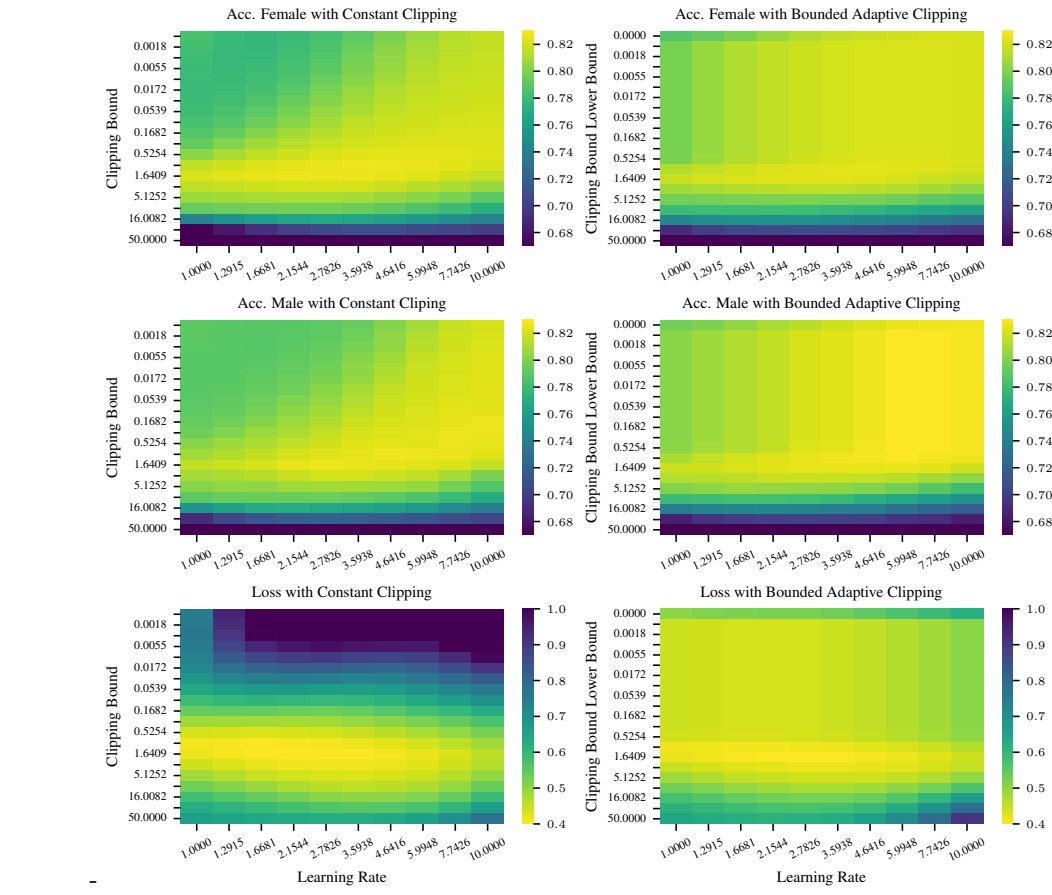

Figure A9: Heatmaps of female accuracy, male accuracy, and loss on Dutch at $\varepsilon = 0.1$ using constant, bounded adaptive, and unbounded adaptive algorithms. The heatmap rows correspond to adaptive clipping with different lower bounds $C_{LB}$; the case $C_{LB} = 0$ (first row) represents unbounded adaptive clipping.

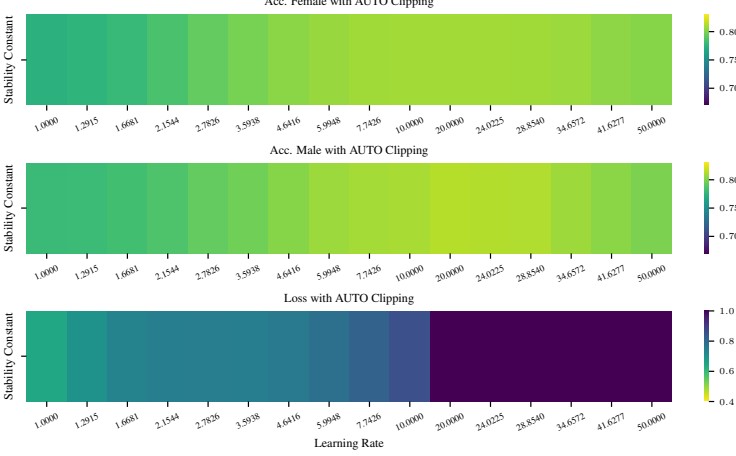

Figure A10: Heatmaps of female accuracy, male accuracy, and loss on Dutch with $\varepsilon = 0.1$ using AUTO. The stability constant is set to the recommended value of $0.01$ Bu et al. (2023).

## C EXTRA RESULTS AND COMPARISONS

### C.1 COMPARISON WITH FAIRDP

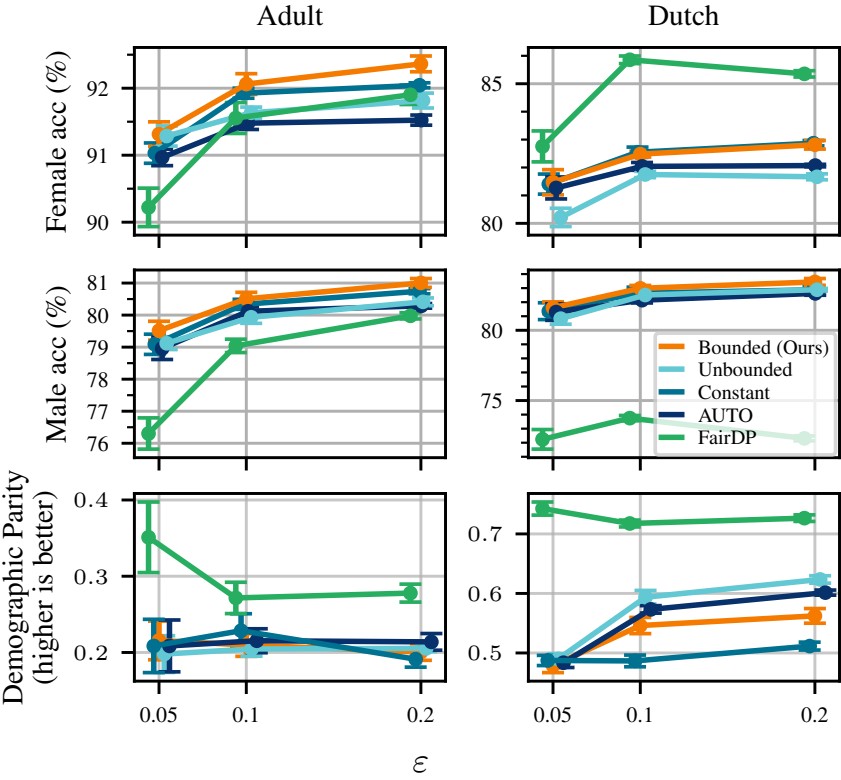

Figure A11: Gender-specific accuracy and demographic parity versus privacy budget on the Adult and Dutch datasets by adding FairDP (Tran et al., 2025). FairDP attains higher demographic parity but substantially lower accuracy relative to the other methods. Results are averaged over 10 seeds, with standard-error bars ($\delta = 10^{-5}$).

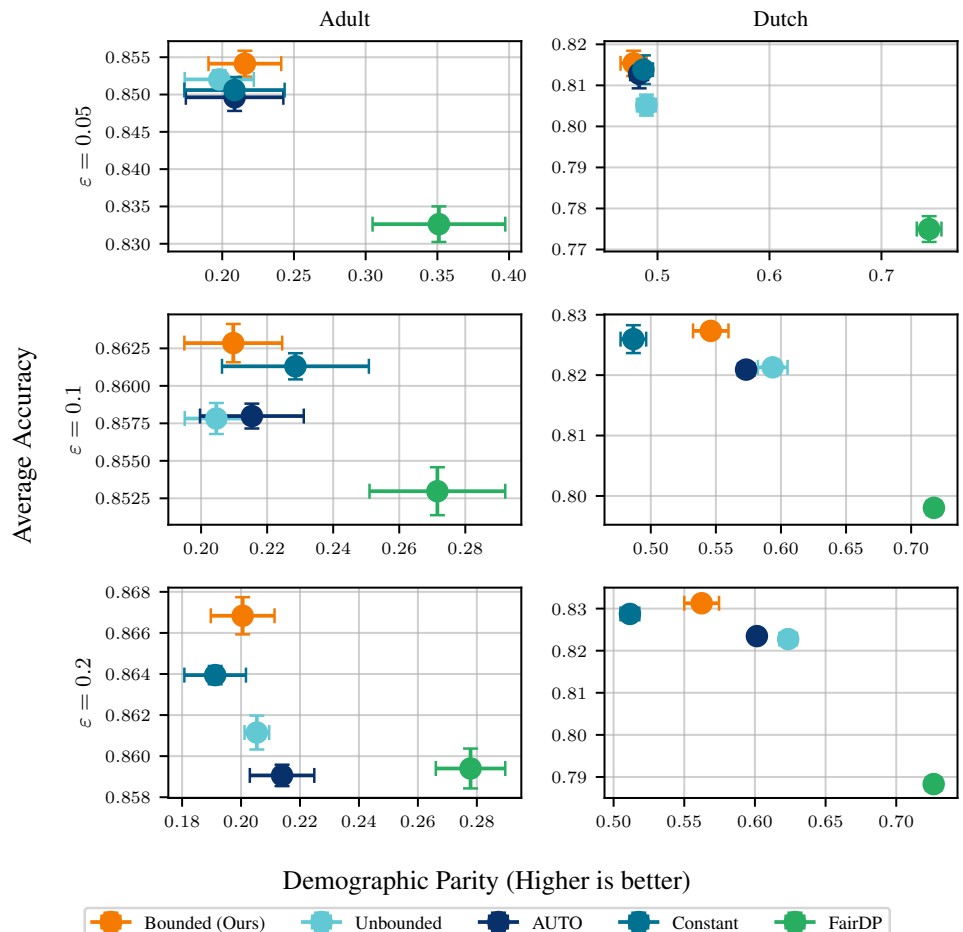

Figure A12: Accuracy versus demographic parity on the Adult and Dutch datasets. Adding from Figure 3b, FairDP (Tran et al., 2025) is included, which achieves fairer results, but worse accuracy. Results are averaged over 10 seeds, and error bars denote standard errors ($\delta = 10^{-5}$).

## C.2 ADAPTIVE CLIPPING WITH A HEURISTIC LOWER BOUND

Increasing the flexibility of adaptive clipping by introducing the lower bound $C_{LB}$ expands the hyperparameter space and thus raises the computational cost of tuning (see Section 3.1). The heatmaps in Appendix B.3 indicate that the hyperparameter landscape along $C_{LB}$ is relatively flat over a broad range, suggesting that many values of $C_{LB}$ lead to similar performance. Motivated by this observation, we consider a simple heuristic choice, $C_{LB} = 0.1$, which eliminates the need for exhaustive tuning along this dimension. While fixing the lower bound may yield a small degradation in optimal accuracy, the heuristic variant remains consistently competitive across architectures and datasets, as shown in the rest of this subsection.

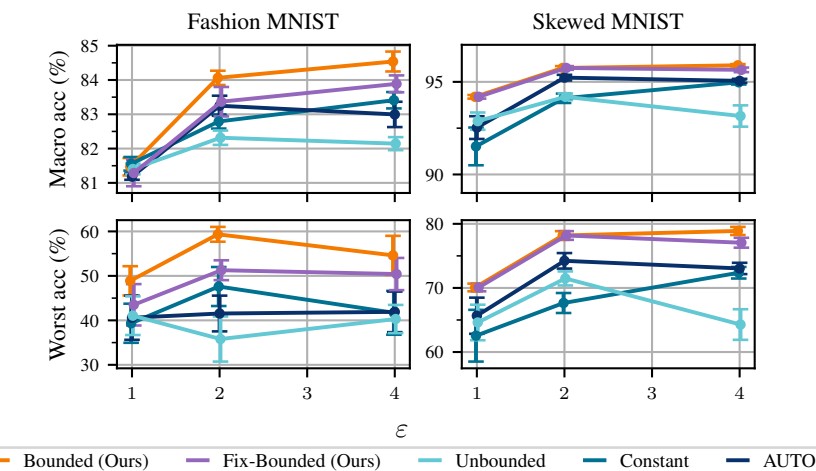

Figure A13: Training a two-layer CNN with DP from scratch. Top: macro accuracy; bottom: worst-class accuracy across privacy budgets, all with optimally tuned hyperparameters. We compare constant clipping, unbounded adaptive clipping, automatic clipping, our bounded adaptive clipping, and the heuristic lower-bound variant ($C_{LB} = 0.1$). Bounded adaptive clipping provides the strongest overall performance and yields pronounced gains in worst-class accuracy, particularly at medium and high privacy budgets. The heuristic lower bound matches the performance of fully tuned bounded clipping while avoiding the additional tuning cost. Curves show means over 10 runs with standard errors ($\delta = 10^{-5}$); points are horizontally jittered for clarity.

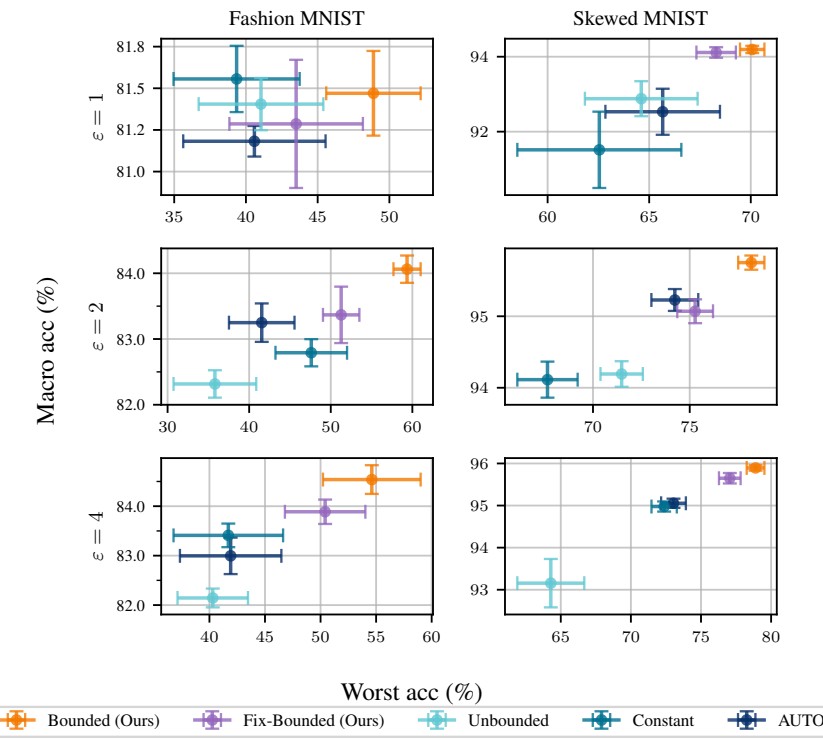

Figure A14: Macro accuracy versus worst-class accuracy for the two-layer CNN across privacy budgets. Each point corresponds to an optimally tuned run under one of the clipping strategies. Bounded adaptive clipping lies on the empirical Pareto frontier, while the heuristic lower-bound variant closely tracks its performance, demonstrating that tuning $C_{LB}$ yields only some additional gains.

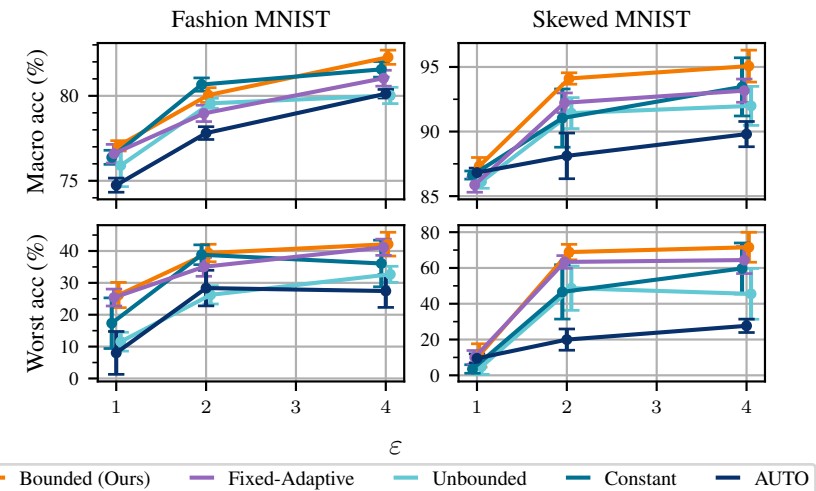

Figure A15: Training ResNet-18 with DP from scratch. Macro accuracy and worst-class accuracy across privacy budgets. We compare constant clipping, unbounded adaptive clipping, automatic clipping, our bounded adaptive clipping, and the heuristic lower-bound variant ($C_{LB} = 0.1$). The trends observed for the smaller CNN model persist at this larger scale: bounded adaptive clipping consistently performs best or near-best, and the heuristic lower-bound variant remains competitive across all privacy levels.

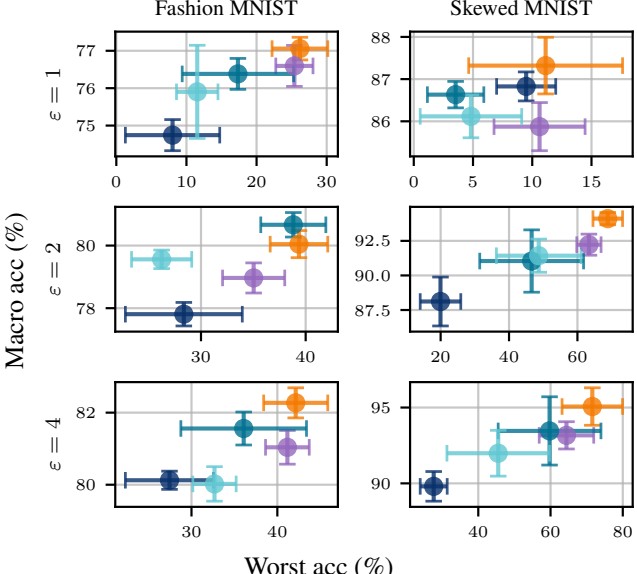

Figure A16: Macro accuracy versus worst-class accuracy for ResNet-18 across privacy budgets, using optimally tuned hyperparameters. Bounded adaptive clipping again aligns with the empirical Pareto frontier, while the heuristic lower-bound method often achieves similar trade-offs with reduced tuning effort.

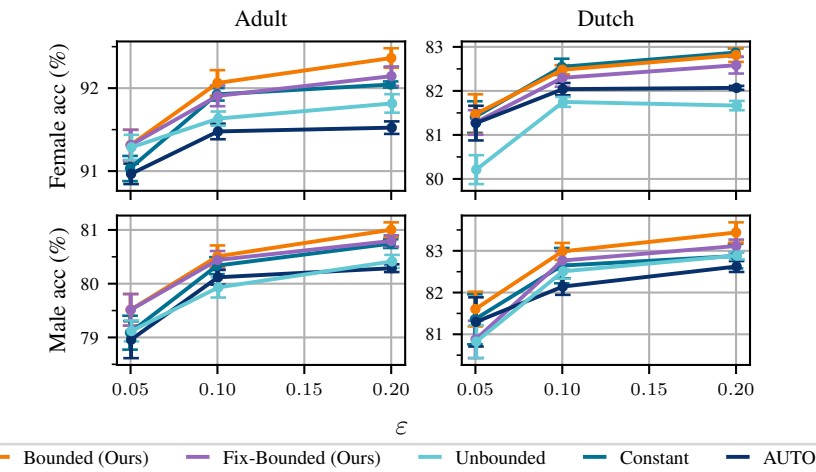

Figure A17: Evaluation about gender-specific accuracy on tabular datasets with the same algorithms in Figure A13. The heuristic lower-bound variant ($C_{LB} = 0.1$) closely matches the performance of fully tuned bounded adaptive clipping for both macro accuracy and worst-class accuracy. These results indicate that fixing $C_{LB}$ provides an effective low-cost alternative for tabular data as well.

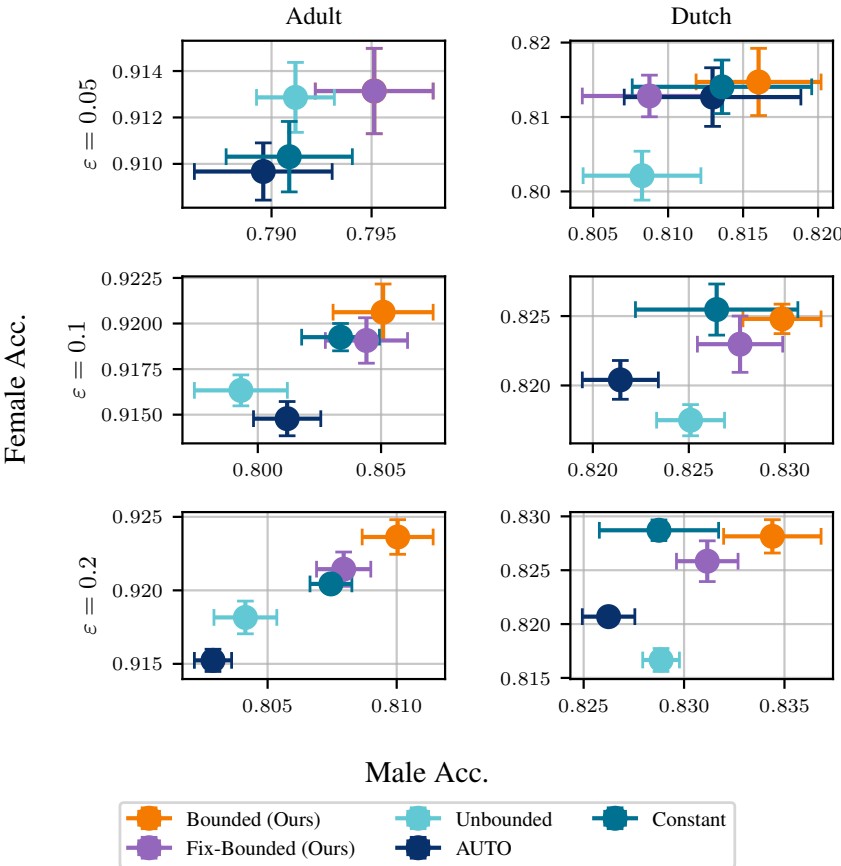

Figure A18: Evaluation about gender-specific accuracy on tabular datasets with the same algorithms in Figure A13. The heuristic lower-bound variant ($C_{LB} = 0.1$) closely matches the performance of fully tuned bounded adaptive clipping for both macro accuracy and worst-class accuracy. These results indicate that fixing $C_{LB}$ provides an effective low-cost alternative for tabular data as well.

