# OpenReview forum: "Mitigating Disparate Impact of Differentially Private Learning through Bounded Adaptive Clipping"
_ICLR.cc/2026/Conference — Submitted to ICLR 2026_

### Official Review · Reviewer_2YPx · 2025-11-01

**Soundness:** 3
**Presentation:** 3
**Contribution:** 3
**Rating:** 8
**Confidence:** 2

**Summary:**

This work proposes lower-bounded adaptive clipping for differential privacy learning to address disparate impact of DP learning on minority and confusable groups. The method leads to improvement in worst-class accuracy for skewed and Fashion MNIST.

**Strengths:**

- The work identifies common issue of disappearing clipping bounds of current works and tackles important problem of ensuring ML fairness for minority groups and shows improvement over SOTA.
- Applied to 4 datasets (Skewed, Fashion MNIST, Adult, Dutch) and 3 architectures ResNet-18, CNN, Logistic Regression
- Testing under Realistic Constraints (DP-HPO)

**Weaknesses:**

- the method adds additional hyperparameters to tune increasing the complexity of the training this is a weakness common to the family of methods
- the paper could be strengthened by providing more explicit guidance or a low-cost heuristic for setting C_LB

**Questions:**

Hyperparameters (Target quantile γ=0.5, Multiplier τ=2.5, and learning rate ηC =0.2) across experiments are fixed, but tuning results in Table A1 show large STD (e.g., τ on Fashion MNIST is 2.7438±3.1248). Could the authors clarify the definition of "stability" used that justified fixing these parameters, despite the high variance observed in preliminary tuning results?

---

> ### Author Response · Authors · 2025-12-03
>
> We thank the reviewer for the constructive feedback and for recognizing the contributions and experimental scope of our work.
>
> > Hyperparameters across experiments are fixed, but tuning results in Table A1 show large STD. Could the authors clarify the definition of "stability" used that justified fixing these parameters, despite the high variance observed in preliminary tuning results?
>
> We clarify our use of the term “stability’’ as “accuracy stability’’ in Appendix A.2 under subsection "The sensitivity of hyperparameters", meaning that model accuracy is only weakly sensitive to variations in the adaptive-clipping hyperparameters.
>
> Although the raw hyperparameter values explored during tuning may vary widely, this variability is not our notion of stability; what matters is that accuracy remains largely unchanged across this range. As shown in Appendix A.2 (Figure A1), broad regions of the $(\tau, \gamma, \eta_C)$ space yield nearly identical accuracy, forming a performance plateau. This robustness of accuracy supports our decision to fix these hyperparameters across experiments.
>
> > the paper could be strengthened by providing more explicit guidance or a low-cost heuristic for setting $C_{LB}$
>
> > the method adds additional hyperparameters to tune increasing the complexity of the training this is a weakness common to the family of methods
>
> We now provide explicit guidance for selecting the lower bound and recommend using a fixed value of $C_{LB} = 0.1$ when computational resources for hyperparameter tuning are limited(see Section 4.1 Methodology, and further discussion and experimental evaluations in Appendix C.2: Adaptive clipping with a heuristic lower bound).
>
> Our results show that tasks with similar input scales and identical architectures tend to share similar optimal $C_{LB}$ values, indicating that this hyperparameter transfers well across related settings. The heatmaps in Appendix B.3 further show that near-optimal worst-class accuracy is achieved when $C_{LB} \approx 0.1$, supporting the use of the proposed heuristic value.

---

### Official Review · Reviewer_ow6d · 2025-11-01

**Soundness:** 2
**Presentation:** 2
**Contribution:** 2
**Rating:** 4
**Confidence:** 5

**Summary:**

This paper addresses a critical issue at the intersection of Differential Privacy (DP) and fairness: the disparate impact of DP-trained models on different demographic groups, particularly minority or challenging subgroups. The core mechanism investigated is adaptive gradient clipping, which is standard practice in training with DP-SGD (Differentially Private Stochastic Gradient Descent).

**Strengths:**

The technical quality appears sound. The proposed BAC method is a direct, mathematically clear modification of existing techniques, making it easy to integrate. The experimental design is robust, contrasting BAC not only with standard DP-SGD but also with existing adaptive clipping schemes (like AUTO), providing a necessary control group to validate the contribution. The reported results clearly show performance gains on fairness metrics (e.g., disparate accuracy), suggesting the method effectively achieves its stated goal.

**Weaknesses:**

- The primary weakness lies in the selection and motivation of the lower bound, $C_{\text{min}}$. While the method's effectiveness hinges on this parameter, the paper does not provide sufficient theoretical guidance for its choice. Currently, $C_{\text{min}}$ appears to be a manually tuned hyperparameter. This reduces the actionability of the insight. If $C_{\text{min}}$ is set too high, it negates the benefits of adaptive clipping; if set too low, it fails to help the minority group. The paper needs a more rigorous study or a heuristic/theoretical justification for how to choose $C_{\text{min}}$ relative to, for instance, the empirical gradient norm distribution of the minority group.
- The paper positions its work as mitigating disparate impact. However, the experiments mainly compare BAC to DP-SGD variations (which are privacy-focused) rather than methods explicitly designed for fairness under DP, such as Group DP-SGD (which uses group-specific clipping thresholds or noise scales) or DP versions of re-weighting or adversarial debiasing. A weakness is the absence of a direct comparison showing how BAC's implicit fairness improvement compares to the explicit fairness control achieved by these alternative methods. Without this, the reader cannot fully assess BAC's place in the fairness-under-DP literature.
- The assessment of disparate impact appears to focus predominantly on Disparate Accuracy (difference in accuracy between groups). In many real-world applications (like loan approval or recidivism prediction), metrics like Equal Opportunity Difference (difference in False Negative Rates, $FNR$) or Predictive Parity (difference in Positive Predictive Values, $PPV$) are often more critical. The experiments are insufficient without evaluating the impact of BAC on these other crucial fairness metrics, which could potentially reveal trade-offs not visible through accuracy alone.

**Questions:**

- Can the authors elaborate on whether $C_{\text{min}}$ can be justified or estimated without extensive hyperparameter search? For instance, could $C_{\text{min}}$ be set to a small percentile (e.g., the $5^{\text{th}}$ percentile) of the historical L2 gradient norms observed on the entire training set, or perhaps only on the minority/underperforming subgroup?
- Does the enforcement of $C_{\text{min}}$ affect the required noise level for a fixed privacy budget $\epsilon$, compared to a standard (unbounded) adaptive clipping method? Intuitively, a bounded clip norm could stabilize the bound variance, but a formal discussion on the impact of BAC on the final noise scale and the $\epsilon$ calculation is needed.

---

> ### Author Response · Authors · 2025-12-03
>
> We thank the reviewer for these detailed questions and for highlighting important aspects of the lower-bound design, privacy accounting, and fairness evaluation. Below we provide clarifications addressing each point.
>
> > Can the authors elaborate on whether $C_min$ can be justified or estimated without extensive hyperparameter search? For instance, could $C_min$ be set to a small percentile (e.g., the $5th$ percentile) of the historical L2 gradient norms observed on the entire training set, or perhaps only on the minority/underperforming subgroup?
>
> > The primary weakness lies in the selection and motivation of the lower bound, $C_{min}$. While the method's effectiveness hinges on this parameter, the paper does not provide sufficient theoretical guidance for its choice. Currently, $C_{min}$ appears to be a manually tuned hyperparameter. This reduces the actionability of the insight. If $C_{min}$ is set too high, it negates the benefits of adaptive clipping; if set too low, it fails to help the minority group. The paper needs a more rigorous study or a heuristic/theoretical justification for how to choose $C_{min}$ relative to, for instance, the empirical gradient norm distribution of the minority group.
>
> We have added a  recommendation to set $C_{\min} = 0.1$ when computing resources for hyperparameter tuning are limited (see Appendix C.2: Adaptive clipping with a heuristic lower bound and Figures A13-A18). As shown in Appendix B.3, tasks with similar input scales and the same architecture tend to share similar optimal $C_{\min}$​ values, indicating that the hyperparameter transfers well across related settings. We also observe that near-optimal worst-class accuracy is consistently achieved when $C_{\min} \approx 0.1$, which motivates the heuristic. Empirically, this allows the adaptive mechanism to follow early training dynamics while still preventing collapse from over-clipping in later stages.
>
> The optimal choice of $⁡C_{\min}$​ depends on factors such as input magnitude, the gradient-norm distribution near convergence, group imbalance, and model architecture. Using a small percentile of historical gradient norms requires extra privacy budget, which will introduce more noise than the current method. Moreover, our current algorithm can encourage the clipping bound to converge at a certain percentile by setting target quantile $\gamma$.
>
> > Does the enforcement of $C_min$ affect the required noise level for a fixed privacy budget $\epsilon$, compared to a standard (unbounded) adaptive clipping method? Intuitively, a bounded clip norm could stabilize the bound variance, but a formal discussion on the impact of BAC on the final noise scale and the $\epsilon$ calculation is needed.
>
> The choice of $C_{\min}$​ does not affect the required noise level for a fixed privacy budget. Our implementation follows normalized DP-SGD [1], where the clipping bound (whether adaptive [2, 3] or our lower-bounded adaptive) is decoupled from the sensitivity of the released gradient. Each per-sample gradient is clipped and rescaled to unit norm before averaging, giving a fixed $l_2$-sensitivity of 1 regardless of the value of $C_t$ or $⁡C_{\min}$​.
>
> Because the sensitivity remains constant, the privacy accountant (PRV or RDP) applies exactly as in standard DP-SGD. The noise multipliers needed to achieve a target $(\varepsilon,\delta)$ are therefore unchanged by the introduction of the lower bound. In other words, BAC stabilizes the clipping dynamics but does not alter the privacy guarantee or noise calibration.
>
> [1]. De et al., Unlocking high-accuracy differentially private image classification through scale, arXiv.2022.
>
> [2]. Andrew, Galen, et al. Differentially private learning with adaptive clipping. NeurIPS.
>
> [3]. Esipova, Maria S., et al. Disparate Impact in Differential Privacy from Gradient Misalignment. ICLR.

---

> > ### Author Response · Authors · 2025-12-03
> >
> > > The paper positions its work as mitigating disparate impact. However, the experiments mainly compare BAC to DP-SGD variations (which are privacy-focused) rather than methods explicitly designed for fairness under DP, such as Group DP-SGD (which uses group-specific clipping thresholds or noise scales) or DP versions of re-weighting or adversarial debiasing. A weakness is the absence of a direct comparison showing how BAC's implicit fairness improvement compares to the explicit fairness control achieved by these alternative methods. Without this, the reader cannot fully assess BAC's place in the fairness-under-DP literature.
> >
> > Our primary disparate impact baseline is the unbounded adaptive clipping method of Esipova et al. [1], which explicitly aims to mitigate disparate impact in DP training. Their work argues that adaptive clipping can correct gradient misalignment and mitigating the disparate-impact problem. Our results show that this mitigation is incomplete: without a lower bound, the adaptive rule can collapse and disproportionately suppress gradients from minority or hard groups. Bounded adaptive clipping addresses this failure mode directly and achieves less disparate impact over this established baselines.
> >
> > To further situate our method within the disparate-impact-under-DP literature, we have now added comparisons with FairDP [2], a representative group-DP-SGD approach. These results, now added in Appendix C.1, show that our method is on the empirical Pareto front; while FairDP can attain higher demographic parity, it does so at the cost of substantially lower accuracy. Including FairDP provides a more complete picture of how bounded adaptive clipping relates to methods that mitigating disparate impact through group-specific clipping thresholds or noise scales.
> >
> > [1]. Esipova, Maria S., et al. Disparate Impact in Differential Privacy from Gradient Misalignment. ICLR.
> >
> > [2]. Tran, Khang, et al. FairDP: Certified Fairness with Differential Privacy. arXiv:2305.16474 (2023).
> >
> > > The assessment of disparate impact appears to focus predominantly on Disparate Accuracy (difference in accuracy between groups). In many real-world applications (like loan approval or recidivism prediction), metrics like Equal Opportunity Difference (difference in False Negative Rates, ) or Predictive Parity (difference in Positive Predictive Values, ) are often more critical. The experiments are insufficient without evaluating the impact of BAC on these other crucial fairness metrics, which could potentially reveal trade-offs not visible through accuracy alone.
> >
> > We focus specifically on the problem of disparate impact, rather than attempting to satisfy multiple fairness notions. Many fairness definitions exist, each motivated by different real-world applications, and it is well understood that most of them cannot be simultaneously satisfied by a single model (see, e.g., [4]).
> >
> > In this work, we use accuracy parity as our primary measure of disparate impact on multi-class datasets, following the prior DP–disparate-impact literature [1, 2]: this metric is widely adopted and serves as a representative measure of disparate impact in DP-SGD. Prior work [3] further shows that improvements in accuracy-based metrics often correlate with improvements in related error-rate-based metrics. Across all experiments, bounded adaptive clipping consistently reduces accuracy disparities, indicating that it mitigates the underlying gradient-suppression bias.
> >
> > However, to also evaluate the possible tradeoffs, we have now added demographic parity for the models trained with tuned hyperparameters (see Figure 4 and the discussions in Section 4.2) on the tabular datasets. The results indicate that our bounded adaptive clipping is on the empirical Pareto front.
> >
> > [1]. Xu, Depeng, et al. Removing disparate impact on model accuracy in differentially private stochastic gradient descent. KDD.
> >
> > [2]. Esipova, Maria S., et al. Disparate Impact in Differential Privacy from Gradient Misalignment. ICLR.
> >
> > [3]. Demelius, Lea, et al. Private and Fair Machine Learning: Revisiting the Disparate Impact of Differentially Private SGD. arXiv preprint arXiv:2510.01744 (2025).
> >
> > [4] Defrance & De Bie: Maximal Combinations of Fairness Definitions, JAIR 2025.

---

### Official Review · Reviewer_Lhr2 · 2025-11-05

**Soundness:** 2
**Presentation:** 3
**Contribution:** 2
**Rating:** 4
**Confidence:** 3

**Summary:**

The paper studies DP-SGD with adaptive clipping (similar to Andrew et al., 2021), which privately tracks a quantile of per-sample gradient norms and updates a global clipping bound so that roughly a target fraction of gradients are clipped. It identifies a failure mode for this method: as training progresses and most gradients shrink, the estimated proportion above the bound drops, the bound keeps shrinking, and can collapse toward zero. This disproportionately hurting minority or hard groups whose gradients remain larger.

To prevent this, the authors add a tunable lower bound $C_{LB}$ on the clipping bound (bounded adaptive clipping): when the adapted bound would fall below $C_{LB}$, they clip at $C_{LB}$ instead. Experiments on image (Fashion-MNIST, Skewed-MNIST) and tabular (Adult, Dutch) datasets show improved worst-class accuracy and competitive macro accuracy versus constant clipping, unbounded adaptive clipping, and AUTO (Bu et al., 2023). Because the proposed method introduces an extra hyperparameter, they also evaluate with DP-HPO and report similar or better performance under the accounted privacy budget.

**Strengths:**

The paper identifies a failure mode of earlier adaptive clipping methods and proposes a simple fix, with experiments demonstrating that it alleviates the issue. It's clearly written.

**Weaknesses:**

- Theorem 3.2 does not provide a precise privacy guarantee. The privacy–accuracy trade-off would be much clearer if the authors specified the resulting $\epsilon$ as an explicit function of $T$, the subsampling rate and the noise multipliers $\sigma_{grad}, \sigma_{count}$. In its current form, the guarantee is hard to interpret.

- While the mean-estimation example is interesting, it seems specific. Is the failure primarily driven by the setup in which the minority group is strictly smaller than the majority? How general is the phenomenon beyond that specific data structure?

- For image data, the “group” is defined by the class label, which isn’t a protected attribute, so the fairness interpretation is unclear.

**Questions:**

- Theorem 3.2 seems to rely on Lemma 3.1, which assumes both Gaussian mechanisms have sensitivity 1. While counting has sensitivity 1, the private gradient mean (after averaging) does not. I assume the privacy amplification by subsampling can also complicates the results. How do you handle sensitivity for this?

- How does proposed method perform when the size of the minority group is similar to that of the majority group?

- How does the method perform in terms of other fairness metrics such as per-group FPR/TPR, gap between group accuracies, etc.?

---

> ### Author Response · Authors · 2025-12-03
>
> We thank the reviewer for the insightful comments and have addressed each point in detail below.
>
> > Theorem 3.2 seems to rely on Lemma 3.1, which assumes both Gaussian mechanisms have sensitivity 1. While counting has sensitivity 1, the private gradient mean (after averaging) does not. I assume the privacy amplification by subsampling can also complicate the results. How do you handle sensitivity for this?
>
> Assuming sensitivity 1 in Lemma 3.1 was wlog. We have now revised Lemma 3.1 to explicitly allow for arbitrary sensitivities for clarity. Our implementation follows normalized DP-SGD [1], where each per-sample gradient is first clipped and then rescaled to at most unit norm (1). Our clipping mechanism therefore ensures the gradient has sensitivity 1. By doing this, we also decouple the learning rate from the clipping bound.
>
> Privacy amplification by subsampling is fully incorporated through the standard privacy accountant (e.g., PRV or RDP) without any additional complications. Noise multipliers $\sigma_{grad}$ and $\sigma_{count}$ are calibrated through the accountant’s standard subsampled-Gaussian mechanism; see the accountant implementation in Opacus (https://github.com/meta-pytorch/opacus/blob/main/opacus/accountants/utils.py).
>
> [1]. De, Soham, et al. Unlocking high-accuracy differentially private image classification through scale. arXiv preprint arXiv:2204.13650 (2022).
>
> > How does proposed method perform when the size of the minority group is similar to that of the majority group?
>
> The effect we study is driven by differences in gradient-norm distributions rather than by relative group sizes. As we demonstrate in the Fashion-MNIST experiments (Fig. 2, A13, A15), when groups are of similar size but differ in difficulty, our method continues to provide clear benefits. In these cases, the harder group tends to produce larger gradients, which are disproportionately suppressed when the clipping bound adapts to the easier subset of the data. Disparate impact therefore arises even with perfectly balanced group sizes, solely because of these differences in gradient norms. In these settings, bounded adaptive clipping improves worst-class accuracy by preventing excessive downweighting of these larger-gradient groups. When groups are of similar size and have comparable difficulty, namely, their gradient norms dynamics are similar, the lower-bound constraint is rarely active, and our bounded adaptive clipping behaves similarly to unbounded adaptive clipping.
>
>
> >For image data, the “group” is defined by the class label, which isn’t a protected attribute, so the fairness interpretation is unclear.
>
> Our study follows well-established prior work [1,2,3] that interprets accuracy parity across classes as a form for evaluating the disparate impact. This metric quantifies disparate impact in prediction quality across groups, whether they correspond to demographic or semantic categories.
>
> Moreover, confusable classes can be viewed as performance-critical subgroups who are more vulnerable to over-clipping and noise under DP-SGD. In this sense, mitigating disparate impact across class labels serves the same purpose as mitigating disparate impact across demographic groups: preventing a subset of the data from being disproportionately harmed by the DP mechanism.
>
> [1]. Xu, Depeng, et al. Removing disparate impact on model accuracy in differentially private stochastic gradient descent. KDD.
>
> [2]. Esipova, Maria S., et al. Disparate Impact in Differential Privacy from Gradient Misalignment. ICLR.
>
> [3]. Demelius, Lea, et al. "Private and Fair Machine Learning: Revisiting the Disparate Impact of Differentially Private SGD." arXiv preprint arXiv:2510.01744 (2025).
>
> > How does the method perform in terms of other fairness metrics such as per-group FPR/TPR, gap between group accuracies, etc.?
>
> In the revised version, we have added results for demographic parity, defined as
>
> $$ \text{Demographic Parity} = \frac{\min_a PR_a}{\max_a PR_a},$$
>
> using tuned hyperparameters on the tabular datasets (see Figure 3b and the discussion in Section 4.2; More experiments are available in Appendix C). The results show that our bounded adaptive clipping lies on the empirical Pareto front.

---

> > ### Author Response · Authors · 2025-12-03
> >
> > > Theorem 3.2 does not provide a precise privacy guarantee. The privacy–accuracy trade-off would be much clearer if the authors specified the resulting  as an explicit function of T, the subsampling rate and the noise multipliers $\sigma_{grad}$ and $\sigma_{count}$. In its current form, the guarantee is hard to interpret.
> >
> > Theorem 3.2 formalizes how the two Gaussian mechanisms:
> >
> > (i) the privatized gradient mean and
> >
> > (ii) the privatized clipping-count update
> >
> > compose within each training step, instead of providing a necessarily loose closed-form $\varepsilon(T,q,\sigma)$. The precise $(\varepsilon,\delta)$ guarantee is obtained numerically by a standard privacy accountant (e.g., PRV or RDP), which takes as input the subsampling rate $q$, number of steps $T$, and the noise multipliers of both mechanisms.
> >
> > Given a target privacy budget, the accountant returns a total noise multiplier which can then be allocated between $\sigma_{\text{grad}}$ and $\sigma_{\text{count}}$ while preserving the overall guarantee. Our theorem therefore clarifies what is being composed, while the accountant provides the numerical privacy-accuracy trade-off as a function of $T$, $q$, $\sigma_{\text{grad}}$​, and $\sigma_{\text{count}}$.
> >
> > We avoid imposing an accountant-specific closed-form to ensure our algorithm can be applied to PRV, RDP, and other accounting methods, as these give tighter accounting than analytical expressions.
> >
> > > While the mean-estimation example is interesting, it seems specific. Is the failure primarily driven by the setup in which the minority group is strictly smaller than the majority? How general is the phenomenon beyond that specific data structure?
> >
> > The failure mode includes but is not limited to the case where one group is smaller. As demonstrated in Fig 1(b), adaptive clipping collapses whenever a subgroup consistently produces larger gradient norms, regardless of the underlying cause. This can arise from sample-size imbalance or from intrinsic difficulty or class similarity, even when groups are perfectly balanced.
> >
> > Our experiments reflect both scenarios: in Fig 2, Skewed-MNIST captures the sample-size imbalance case, while Fashion-MNIST shows that collapse also occurs in balanced settings. This phenomenon can also be found in Figs A13, A15: the worst-performing class accuracy is significantly lower than the macro accuracy. These forms of heterogeneity are common in practical training pipelines, making the phenomenon general rather than example-specific.

---

### Official Review · Reviewer_ABd6 · 2025-11-08

**Soundness:** 3
**Presentation:** 3
**Contribution:** 3
**Rating:** 4
**Confidence:** 5

**Summary:**

This paper identifies a critical issue in differentially private (DP) learning: adaptive clipping methods can lead to vanishing clipping bounds, which disproportionately harm minority or challenging classes. The authors propose a simple yet effective solution—introducing a lower bound on the clipping threshold—and demonstrate its efficacy across multiple datasets and models. The work is well-motivated, methodologically sound, and thoroughly evaluated. It addresses an important problem at the intersection of privacy and fairness, with practical implications for real-world DP training. Experiments across MNIST, Adult and Dutch, show improved worst-class and subgroup accuracy, with competitive macro accuracy, under both optimal hyperparameters and DP-HPO.

**Strengths:**

1.	This manuscript first identifies a well-identified problem, via a failure mode of existing adaptive clipping methods, where clipping bounds collapse during training, leading to unfair outcomes. The toy example in Figure 1 is particularly effective in illustrating this issue.
2.	Authors propose a simple and effective Solution, i.e. DP-HPO. Its bounded adaptive clipping is easy to implement, requires minimal modification to existing DP-SGD pipelines, and comes with a clear privacy guarantee.
3.	The privacy analysis is rigorous, leveraging Gaussian DP composition to account for both gradient and clipping-bound updates.
4.	The paper provides extensive details on hyperparameters, datasets, and experimental setups.

**Weaknesses:**

1.	The DP-HPO introduce a new hyperparameter i.e. the lower-bound of adaptive clipping bound C_LB. The paper shows robustness, but provides limited guidance on principled selection., this could be a practical barrier.
2.	The paper has a limited theoretical analysis about fairness. While motivated by fairness, the paper does not provide a theoretical analysis of how bounded clipping improves fairness guarantees (e.g., in terms of fairness definitions like equalized odds or demographic parity).
3.	The paper compares to AUTO and constant/unbounded clipping. It should provide comparisons with other fairness-oriented DP methods (e.g., DP-SGD-Fair by Xu et al., 2021, FairDP by Liu et al. 2022).

**Questions:**

1.	Can the authors provide a simple theoretical intuition or bound on how the lower bound mitigates disparate impact?
2.	Could the authors provide a sensitivity analysis or a heuristic for setting C_LB?
3.	Have the authors considered evaluating other fairness metrics (e.g., demographic parity, equal opportunity) beyond accuracy parity?
4.	DP-HPO is proposed based on normalized DP-SGD (De et al., 2022). How does the proposed adaptive clipping bound C_LB working with SGD, affect the fairness? The related work shows DP-SGD has the fairness problem. But, Lemma 3.1 and Theorem3.2 are both about privacy.
5.	I admit that The proposed DP-HPO is a simple and effective method, but it seem a little incremental novelty, via introduce the C_LB.

---

> ### Author Response · Authors · 2025-12-03
>
> We thank the reviewer for these detailed and thoughtful questions. Below we provide clarifications regarding the theoretical intuition, sensitivity analysis, fairness/disparate-impact metrics, the interaction with DP-HPO, and the novelty of the proposed lower bound.
>
> > Can the authors provide a simple theoretical intuition or bound on how the lower bound mitigates disparate impact?
> We demonstrate a simple intuitive reason for the mitigation effect in Fig 1 on a toy mean estimation problem (see Section 3.2 for formal statement of the problem):
>
> The core mechanism is that unbounded adaptive clipping tends to continually shrink the clipping threshold as gradients from the majority/easy to fit samples decrease over training. Once the bound becomes too small, gradients from harder or minority groups are uniformly clipped, preventing them from influencing the updates. This creates a form of “majority vote” and leads to disparate impact.
>
> > Could the authors provide a sensitivity analysis or a heuristic for setting C_LB?
> We now provide a concrete heuristic for choosing $C_{LB}$ in the revised version (see Appendix C.2). When hyperparameter tuning is limited, we recommend using a fixed value of $C_{LB}=0.1$ based on our extensive experimental results. The revised version also includes new experiments (Figures A13–A18) showing that this choice achieves competitive accuracy and worst-class performance across datasets and architectures.
>
> To further support this recommendation, Appendix B.3 provides a two-dimensional sensitivity analysis over $C_{LB}$ and batch size. The results show that models with similar input scale and architecture share similar optimal regions, and that near-optimal worst-class accuracy is consistently achieved when $C_{LB}$ lies in a stable range. These findings demonstrate that a small fixed lower bound (e.g., $C_{LB}=0.1$) performs reliably without requiring extensive hyperparameter search.
>
> > Have the authors considered evaluating other fairness metrics (e.g., demographic parity, equal opportunity) beyond accuracy parity?
>
> There exist many fairness definitions and it is well-known that only a few can be satisfied by any single model (see, e.g., [5]). We focus on accuracy parity for the multi-class datasets following prior DP-disparate-impact literature [1, 2, 3]: this metric is widely adopted and serves as a representative measure of disparate impact in DP-SGD. Prior work [4] further shows that improvements in accuracy-based disparate-impact metrics often correlate with improvements in related error-rate-based metrics.
>
> Moreover, to demonstrate the potential tradeoffs betweem fairness metrics and disparate-impact metrics, we have now added results with demographic parity on the tabular datasets for the models trained with tuned hyperparameters (see Figure 3b and the discussion in Section 4.2, more results can be found Figure A12 in Appendix). The results indicate that our bounded adaptive clipping is on the empirical Pareto front.
>
> [1]. Xu, Depeng, et al. Removing disparate impact on model accuracy in differentially private stochastic gradient descent. KDD.
>
> [2]. Esipova, Maria S., et al. "Disparate Impact in Differential Privacy from Gradient Misalignment." ICLR.
>
> [3]. Demelius, Lea, et al. Private and Fair Machine Learning: Revisiting the Disparate Impact of Differentially Private SGD. arXiv:2510.01744 (2025).
>
> [4] Lea Demelius et al., Private and Fair Machine Learning: Revisiting the Disparate Impact of Differentially Private SGD, TMLR 2025.
>
> [5] Defrance & De Bie: Maximal Combinations of Fairness Definitions, JAIR 2025.
>
>
> > DP-HPO is proposed based on normalized DP-SGD (De et al., 2022). How does the proposed adaptive clipping bound C_LB working with SGD, affect the fairness? The related work shows DP-SGD has the fairness problem. But, Lemma 3.1 and Theorem 3.2 are both about privacy.
>
> We adopt the DP-HPO framework from Papernot et al. [1] to evaluate the robustness of our method. The DP-HPO procedure is independent of the specific training algorithm and can be applied to any DP optimization setting, including ours.
>
> The prior work suggested that the clipping in the DP-SGD is the main reason for the disparate impact. Thus, the (unbounded) adaptive algorithm [2] has been proposed as a mitigation. However, we demonstrate that the existing methods have a common failure mode, namely, when the clipping bound shrinks to a small value, the disparate impacts are amplified instead of mitigated. We then show how to fix this issue using our bounded adaptive clipping method.
>
> Lemma 3.1 and Theorem 3.2 prove that our proposed method is still DP, and, that the standard DP accounting methods, e.g., PRV and RDP, can be employed with our algorithm.
>
> [1]. Papernot, N. and Steinke, T. Hyperparameter tuning with renyi differential privacy. ICLR.
>
> [2]. Esipova, Maria S., et al. Disparate Impact in Differential Privacy from Gradient Misalignment. ICLR.

---

> > ### Author Response · Authors · 2025-12-03
> >
> > > I admit that The proposed DP-HPO is a simple and effective method, but it seem a little incremental novelty, via introduce the C_LB.
> >
> > Our proposed method is simple to implement, but the novelty of our work does not lie in algorithmic complexity. Instead, our main contribution is to identify and analyze a common previously unrecognized failure mode of adaptive clipping: when unconstrained, the clipping bound can collapse during training, disproportionately suppressing the gradients of minority or hard subgroups (Sec. 3.2 and Fig. 1). This phenomenon is general, arises across architectures and datasets, and has not been noticed nor addressed in prior DP or DP-disparate-impact literature.
> >
> > In this sense, the simplicity of our novel method is an advantage rather than a limitation: it directly resolves a fundamental issue in adaptive clipping while remaining easy to integrate into existing DP-SGD pipelines.

---

### Official Review · Reviewer_aMme · 2025-11-09

**Soundness:** 2
**Presentation:** 2
**Contribution:** 2
**Rating:** 2
**Confidence:** 4

**Summary:**

This paper diagnoses a failure mode in DP training. It claims that existing adaptive clipping methods cause disparate impact by shrinking the clipping bound to "tiny values" to accommodate the majority group. This, in turn, suppresses the larger gradients from minority or "challenging" samples, harming their performance.
The authors propose "bounded adaptive clipping" as a solution. This method is a minor modification that introduces a tunable hyperparameter, which acts as a floor, preventing the clipping bound from collapsing to zero. The paper shows this simple fix improves worst-class accuracy on skewed image datasets.

**Strengths:**

1. This paper focuses on an important part of DP training.
2. The paper clearly identifies and illustrates a failure mode for unbounded adaptive clipping, where the bound collapses and ignores minorities.

**Weaknesses:**

1. The novel part of this paper is the max() function. This is a minor heuristic, not a new framework.
2. Baseline is not well selected. Why pick the auto clipping? My understanding is that auto clipping is good for hyperparameter tuning since it does not require for clip bound. Why do you want to compare your proposed method with them? I think De et al.(https://arxiv.org/pdf/2204.13650) may be a good choice. They achieve good performance on many datasets. If your method plus theirs can achieve new SoTA results on CIFAR-10 or CIFAR-100 dataset will make your method more stronger.
3. The datasets are toy datasets. I know for a DP paper, it may not be easy for training with ImageNet but at least use CIFAR-10/100.
4. The improvements are not consistent.  Sometimes the proposed method is better than baseline for eps=1 and 4, sometimes it is better for eps=2. Could authors provide more explanation for this? Some improvements are limited.

**Questions:**

See weaknesses.

---

> ### Author Response · Authors · 2025-12-03
>
> We thank the reviewer for the detailed feedback. Below we address the concerns about novelty, baseline selection, dataset choice, and the consistency of improvements.
>
> > The novel part of this paper is the max() function. This is a minor heuristic, not a new framework.
> Our contribution is not centered on algorithmic complexity. The novelty lies first in formally identifying and analyzing a previously unrecognized common failure mode of adaptive clipping: unbounded adaptive clipping can collapse the clipping bound during training and disproportionately suppress gradients from minority or hard subgroups (Sec. 3.2, Fig. 1).
>
> We demonstrate that this failure mode is general, not dataset-specific, and it has not been identified nor addressed in prior DP or DP-disparate-impact literature. The proposed lower bound directly resolves this issue and yields consistent disparate-impact and utility improvements in challenging DP settings. In this sense, the simplicity of the fix is an advantage: it makes the method easy to adopt in real-world DP training pipelines while addressing a fundamental problem in adaptive clipping.
>
> > Baseline is not well selected. Why pick the auto clipping? My understanding is that auto clipping is good for hyperparameter tuning since it does not require for clip bound. Why do you want to compare your proposed method with them? I think De et al.(https://arxiv.org/pdf/2204.13650) may be a good choice. They achieve good performance on many datasets. If your method plus theirs can achieve new SoTA results on CIFAR-10 or CIFAR-100 dataset will make your method more stronger.
>
> De et al. (2022) correspond directly to the “constant clipping’’ baseline in our work, which we already include in all experiments. Their method fixes a global clipping threshold, and its behaviour in our setting is therefore captured by the constant-clipping results reported throughout the paper.
>
> AutoClip is a relevant baseline because, in practice, it behaves like using an extremely small clipping bound: its automatically chosen bound is smaller than nearly all gradient norms. This makes AutoClip a natural comparison point for our study, as it directly illustrates the mechanism we investigate, namely, how overly small clipping bounds suppress minority or hard-group gradients and induce disparate impact.
>
> > The datasets are toy datasets. I know for a DP paper, it may not be easy for training with ImageNet but at least use CIFAR-10/100.
> Our chosen datasets are standard benchmarks for evaluating disparate impact under differential privacy and are widely used in prior work [1, 2, 3]. Skewed MNIST is designed to study the impact on minority groups; Fashion-MNIST contains naturally confusable classes; and Adult and Dutch are tabular datasets with demographic disparities (see Figure 3b and the discussions in Section 4.2).
>
> To the best of our knowledge, disparate-impact-focused DP studies on CIFAR-10/100 are rare, and no established protocols exist for defining group structure or evaluating disparate impact on these datasets. This limits the comparability of disparate-impact analyses on CIFAR-10/100 with existing literature. For this reason, we follow the standard disparate-impact-under-DP benchmarks to ensure meaningful comparisons and interpretable results.
>
>
> [1]. Eugene Bagdasaryan, et al. Differential Privacy Has Disparate Impact on Model Accuracy. NeurIPS.
>
> [2]. Xu, Depeng, et al. Removing disparate impact on model accuracy in differentially private stochastic gradient descent.ACM SIGKDD.
>
> [3]. Esipova, Maria S., et al. Disparate Impact in Differential Privacy from Gradient Misalignment. ICLR.
>
> > The improvements are not consistent. Sometimes the proposed method is better than baseline for eps=1 and 4, sometimes it is better for eps=2. Could authors provide more explanation for this? Some improvements are limited.
>
> Our objective is to improve worst-class (or worst-group) accuracy while maintaining macro accuracy, rather than maximizing macro accuracy alone. Consequently, the most relevant metric in Figure A13, A15 is the bottom curve (worst-class accuracy), not the top one (macro acc). We have added experiments with demographic disparities in Figure 3b, where our algorithm is consistently on the Pareto frontier.
>
> Moreover, many of the small differences in macro accuracy across $\epsilon$ values are covered by the 1 standard error bars in our figures, indicating that the apparent fluctuations are likely due to random variability rather than systematic differences between the methods. This is typical in settings where multiple clipping schemes achieve similar overall utility, and demonstrates that our method is on par with the baselines on macro accuracy.
>
> In contrast, the improvements in worst-class accuracy are very consistent and likely not due to chance alone across all settings: our bounded adaptive clipping therefore reliably improves the performance of minority or confusable classes without degrading overall accuracy.

---

### Meta-Review · Area_Chair_nWap · 2025-12-30

**Summary:**

The paper identifies that current adaptive clipping methods might lead to extremely small gradients, which might harm underrepresented groups. To address this, it introduces a lower bound. The reviewers express the concern that both the theoretical contribution of the paper is weak and that the experimentation is limited and conducted on toy datasets. While the first concern seems hard to address in a rebuttal in the first place, the second concern was not addressed (no additional experiments on larger models and datasets) and rather argued away by pointing to prior work also operating on MNIST-style datasets, mentioning that CIFAR-based experiments are uncommon, yet ignoring the established works on DP+fairness that operate, among others on complex face images like FairFace or CelebA, and evaluating sensitive attributes (e.g. gender, race), rather than an MNIST-style class label. Therefore, the AC is confident that the paper would require some major updates to be ready for acceptance.

**Reviewer Concerns:**

The main concerns that have not or only partially addressed can be detailed as follows:

1. The theoretical contribution of adding a lower bound is marginal: which is hard to argue with in the first place. The authors argue that the simplicity is a feature of the method. However, given that this is paired with the simplified privacy analysis, a simplified model of fairness, and weak experimentation (see next points), this might not be sufficient for acceptance.
2. The method adds another hyperparameter, which one then needs to optimize for. While this does not seem a major limitation, and the authors suggest using small values, it adds some additional overhead.
3. The method uses a limited model of fairness. While the rebuttal did a good job specifying which fairness notion was considered (demographic parity), the „theoretical analysis about fairness“ as required by the reviewer (more formal than what was provided) is missing.
4. The privacy analysis might too simplistic, leaving more complex mechanisms, like subsampling out of consideration.
5. The experimentation is performed on toy models and datasets. While the authors argue that prior work did alike, to improve the paper for future submissions, I recommend extending to larger datasets (e.g. FairFace, CelebA), larger models, and considering fairness beyond labels but for other attributes.

**Reviewer Scores:**

Based on the provided rebuttal, it seems unlikely that the reviewers would have raised their scores.

---

### Decision · Program_Chairs · 2026-01-26

Reject